# Bits That Count: Quantifying and Predicting Capabilities of Language Models

**Elizabeth Donoway** [1 2]   **Hailey Joren** [2]   **Michael R. DeWeese** [1]   **Ethan Perez** [2]   **John Schulman** [3]   **Fabien Roger** [2]   **Jan Leike** [2]

## Abstract

When does learning elicit *existing* knowledge, and when does it primarily teach *new* capabilities? We find that the amount of generalizable information language models learn during training predicts the origins of their emergent capabilities. Minuscule amounts of information—in many cases, a few bits in a single example—can unlock large fractions of models' maximum performance when capabilities are *elicited* rather than *taught*. We quantify these learning regimes using excess description length (EDL), an information-theoretic measure of generalizable structure learned from data during training. We find that elicitation and teaching exhibit distinct EDL signatures that characterize the predominant learning process as information scales: elicitation requires orders of magnitude less information than teaching to comparable performance. We demonstrate that EDL provides a practical tool for quantitatively estimating the maximum amount of predictive information models can compress from data into trainable parameters during learning. These capacity limits describe optimal tradeoffs between data and parameter count that robustly predict when parameter-efficient fine-tuning methods (*e.g.*, LoRA) will underperform full fine-tuning.

## 1. Introduction

Large language models (LLMs) acquire a vast and diverse set of capabilities during pretraining (Wei et al., 2022), many of which remain latent until surfaced through post-training interventions such as supervised fine-tuning (SFT) or reinforcement learning from human feedback (RLHF). Elicitation is the process of drawing out these pre-existing capa-

bilities (whether they are latent or already demonstrated), whereas teaching endows a model with capabilities it lacks. Though these two learning regimes differ, standard evaluation metrics fail to distinguish them: both can start from identical zero-shot performance, and both can improve with training to achieve identical final performance.

This distinction matters practically and for safety. If a dangerous capability is latent, it may be elicited with minimal training data or via adversarial attacks (Yang et al., 2023; Wei et al., 2023; Zou et al., 2023; Liu et al., 2024), or it may be accidentally elicited in novel situations that evaluations may not have accounted for, either due to accidental alignment drift (Qi et al., 2023) or due to misalignment (van der Weij et al., 2024). If it must be taught, the information barrier is higher and more predictable. Current evaluation practices risk conflating these cases: a fine-tuning-based evaluation that inadvertently teaches new capabilities may overstate or misidentify what could be elicited from a deployed model.

To make this abstract distinction concrete, consider two scenarios that result in the same final performance on a task:

1. **Elicitation:** A model achieves high, generalizable performance by training for many epochs on a single, unique example. This is possible only if the model can recognize a generalizable pattern it already knows within that instance, using the example as a tool for accessing a latent, preexisting capability, rather than treating it as a fact to memorize.
2. **Teaching:** A model requires a diverse set of many unique examples to achieve the same performance. This implies the underlying pattern is unknown to the model, and it must see numerous variations to construct a new, useful representation of the concept.

Though the final outcomes are the same, the latter requires substantially more information to be learned by the model.

Adopting the perspective of learning as compression, we use excess description length (EDL) (Donoway et al., 2026; Donoway, 2026), an information-theoretic metric that quantifies the predictive information compressed into model parameters during training (Figure 1). EDL measures the reduction in codelength achieved through learning—the "bits

[1]Department of Physics, University of California, Berkeley, CA USA [2]Anthropic, San Francisco, CA USA [3]Thinking Machines, San Francisco, CA USA. Correspondence to: Elizabeth Donoway <donoway@berkeley.edu>.

*Proceedings of the $43^{rd}$ International Conference on Machine Learning*, Seoul, South Korea. PMLR 306, 2026. Copyright 2026 by the author(s).

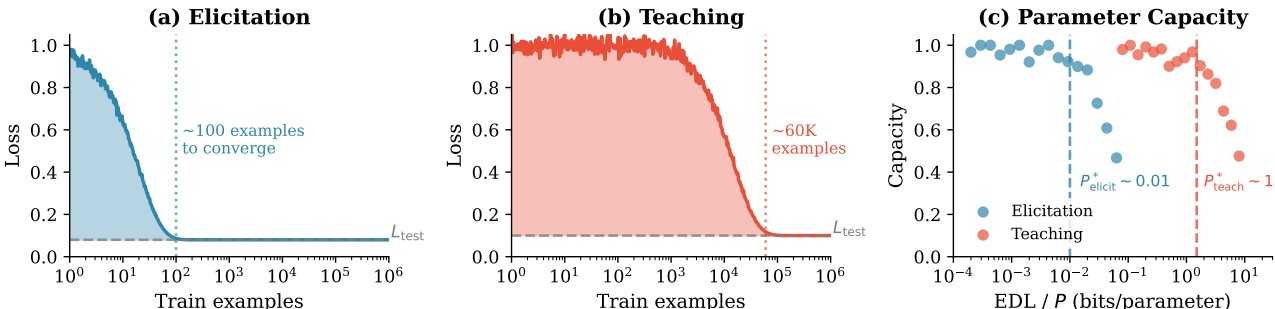

*Figure 1.* **Teaching requires learning significantly more information than elicitation.** Training diagrams of (a) eliciting latent capabilities, (b) teaching new capabilities, and (c) parameter capacity, comparing how elicitation learns generalizable structure (achieves low generalization error) with significantly less information than teaching. The dataset's minimum description length (MDL) is the total area under each training curve, representing the number of bits provided to the model (through the labels of the training dataset), as quantified by the model itself. The final converged test loss $L_{\text{test}}$ (horizontal dashed lines) represents the model's generalization error after training. The excess description length (EDL) is the shaded area between the training curve and generalization error, which represents the generalizable information that is absorbed by the model's parameters during training. EDL quantifies the expected reduction in label stream codelength achieved through the learning process.

that count" toward generalization rather than memorization. We use this approach to establish scaling trends for model elicitation and teaching, revealing that a model's preexisting capabilities are well-predicted by how efficiently it extracts general patterns from its training data. We propose EDL as an operational metric for distinguishing elicitation from teaching and for predicting parameter capacity constraints in fine-tuning.

EDL provides several practical tools for understanding learning dynamics:

- The scaling of EDL per token with dataset size exhibits qualitatively different signatures that characterize the *predominant* learning process at each scale (section 4). Elicitation is associated with decreasing returns to learned structure as data scale, whereas scaling dataset size improves the data efficiency of teaching predictive structure.

- Causal interventions confirm that EDL signatures reflect the distinction between latent versus absent capability (section 5). Pre-training a model on a related skill, thereby making the target capability latent, shifts the EDL scaling from teaching-like to elicitation-like behavior.

- EDL predicts when parameter-efficient fine-tuning methods like LoRA become capacity-constrained (section 6). When the task requires storing more bits than the adapter can accommodate, performance and learning degrade relative to full fine-tuning. We find that this capacity threshold differs substantially between regimes: elicitation saturates around 0.01–0.1 bits per trainable parameter or fewer, while teaching requires roughly 1 bit per parameter or more.

Across three model families and several tasks that span multiple domains and formats, we find that if capabilities are present but latent, a very small amount of information (in the form of trainable parameters or examples) suffices to recover large fractions of a model's performance gap between its zero-shot baseline and full fine-tuning. In multiple models of varying sizes and architectures, we observe several instances where fine-tuning on a single *randomly sampled* example recovers over 50% of the model's full performance gap and improves performance by 20–95 percentage points (Table 8).

We find that fine-tuning few (many) parameters or using few (many) unique examples are functionally equivalent in terms of the amount of generalizable information a model can learn from the training data, regardless of the model's initial capability level or demonstrated task proficiency. For matched accuracy, models with latent capabilities require and absorb significantly less information than models which must learn entirely new skills.

Our main contributions are as follows:

1. We propose EDL as an operational metric computable from standard training logs and demonstrate that it cleanly distinguishes teaching from elicitation across Llama, Qwen, and custom pre-trained models ranging from 1B to 32B parameters.

2. We provide causal evidence through pre-training interventions that shift learning process and regime signatures.

3. We establish that EDL per parameter predicts optimal data–parameter count tradeoffs by estimating parameter capacity limits, with thresholds varying by one to two orders of magnitude between regimes.

## 2. Method: Excess Description Length

This section defines EDL operationally, including how to compute it from training logs, following the procedure of Donoway et al. (2026). Additional details, explanations, and visualizations are provided in Appendix A.

### 2.1. Setup and Loss Convention

Consider fine-tuning a pretrained model on dataset $D = \{(x_i, y_i)\}_{i=1}^n$, where $x_i$ is an input (*e.g.*, a question or prompt) and $y_i$ is the corresponding label (*e.g.*, an answer). Let $\theta_0$ denote the pretrained parameters and $A$ the training algorithm (optimizer, learning rate, hyperparameters, etc.).

We compute cross-entropy loss only on designated label tokens, typically the answer portion of the output (excluding the prompt and formatting). This convention helps to isolate capability-relevant learning from format acquisition, which can otherwise dominate early training dynamics.

### 2.2. Prequential Minimum Description Length

The prequential minimum description length (MDL) accumulates the cross-entropy loss incurred on each training example before the model has updated on it. Let $\theta_{i-1}$ denote the parameters just before training on example $i$. The MDL is:

$$\mathrm{MDL}(D; \theta_0, A) = \sum_{i=1}^n \ell(\theta_{i-1}; x_i, y_i), \quad (1)$$

where $\ell(\theta; x, y) = -\log p_\theta(y \mid x)$ is the cross-entropy (log-loss) on label tokens.

In practice, training proceeds by batches. For each batch, we accumulate the total log-loss (summed over all labels in the batch) before the gradient update. The sum across all labels in the dataset (*i.e.*, in the first epoch[1]) gives the MDL.

Prequential MDL measures how many bits are needed to encode the train set labels as predictions of the updating model. Labels the model already predicts well require few bits; surprising labels require many.

### 2.3. Excess Description Length

After training to some termination condition (*e.g.*, convergence, some number of epochs, or a fixed compute budget), obtain final parameters $\theta^*$. Evaluate the test loss on held-out

data from the same distribution:

$$L_{\mathrm{test}}(\theta^*) = \frac{1}{n_{\mathrm{test}}} \sum_{j=1}^{n_{\mathrm{test}}} \ell(\theta^*; x_j, y_j). \quad (2)$$

The excess description length is:

$$\mathrm{EDL}(D; \theta_0, A) = \mathrm{MDL}(D; \theta_0, A) - n \cdot L_{\mathrm{test}}(\theta^*). \quad (3)$$

EDL measures the gap between the bits spent encoding training labels during learning and the bits expected when using the final model. This gap represents predictive information compressed into the parameters—structure extracted from the train set that improves generalization on unseen data.

By construction, EDL explicitly (i) separates *data* (information source) from *computation* (extraction mechanism) and (ii) distinguishes *generalization* from *memorization* by excluding the generalization gap (Figure 6). EDL remains negligible when no generalizable structure exists, regardless of training compute and data memorization (Appendix I.10).

### 2.4. Conventions and Normalizations

In addition to raw EDL, we report EDL in several normalized forms, depending on the analysis:

- **EDL per token** ($\mathrm{EDL}/D$, where $D$ is total label token count): Used for scaling analysis. This measures the absorbed information per unit of supervision.

- **EDL per parameter** ($\mathrm{EDL}/P$, where $P$ is trainable parameter count): Used for capacity analysis. This measures how densely information is packed into the adapter.

- **Capacity, or EDL ratio** ($\mathrm{EDL}(P, D)/\mathrm{EDL}_{\mathrm{ref}}(D)$): Used for comparing LoRA to full fine-tuning. The reference EDL is obtained from full fine-tuning or high-rank LoRA on the same data.[2]

### 2.5. Practical Considerations

Several implementation details affect EDL measurement. First, token masking procedures (for the loss computation) should be consistent between the train and test sets. Second, for multi-epoch training, MDL is computed only on the first epoch (first exposure to the data), while test loss is computed using the final trained model (arbitrary epochs). This is because MDL describes the total information content of the source (data), whereas training (computation) serves as a mechanism to extract this information. Third, cross-entropy loss values are typically in nats; in the figures of this paper, we express EDL in bits, by dividing by $\ln 2$.

---

[1]Prequential MDL accumulates loss over the first epoch only, as all labels have been encoded a single time at the end of the first epoch. Accumulation over additional epochs would correspond to encoding redundant information, which would make the description length of the data no longer minimal.

[2]Since LoRA can have different training dynamics than full fine-tuning, we use rank 512 LoRA as a reference, $\mathrm{EDL}_{\mathrm{ref}} = \mathrm{EDL}_{r=512}$, which obtains the same performance as full fine-tuning in all of our experiments.

# 3. Experimental Setup

## 3.1. Models

We study models spanning two orders of magnitude in parameter count, chosen to represent different capability profiles (Appendix C). Llama 3.2 1B, 3B and Llama 3.1 8B are pretrained on large, diverse corpora and have a broad range of capabilities that can be elicited with fine-tuning or prompting. TinyStories–1B uses the Llama 3.2 1B architecture but is pretrained exclusively on simple English text from the TinyStories-v2 corpus (Eldan & Li, 2023), resulting in basic language modeling capability without any specialized knowledge. Qwen2.5 1.5B, 14B, and 32B are pretrained on large corpora primarily composed of examples that provide a foundation for instilling common sense, expert technical knowledge, and reasoning capabilities.

## 3.2. Tasks

Arithmetic tasks use subsets of the DeepMind Mathematics dataset that differ in difficulty, providing natural intervention targets. Chain-of-thought reasoning tasks use DeepSeek R1-generated mathematical, scientific, and technical problem solving examples (MATH500, AIME-24, GPQA-Diamond, word games, crossword puzzles) to assess extended reasoning capability. Additional tasks include simple English language modeling (TinyStories-v2), instruction following (Alpaca), reading comprehension (BoolQ), and high school science (ARC–Easy/Challenge). We validate elicitation classification using independent baselines: multi-shot prompting, as well as logit bias correction for multiple-choice tasks, both of which yield significant performance improvements for models we classify as being in the elicitation regime (Appendix D). Details are provided in Appendix B.

## 3.3. Training Configuration

For parameter-efficient fine-tuning, we use LoRA applied to all layers. We train ranks 1 to 512, including uniformly randomly sampled sparse subsets of parameters within rank 1. Trainable parameter counts range from a single parameter to hundreds of millions depending on model size and rank. Full training details are provided in Appendices G and H.

For pre-training interventions, we first train on the intervention task (*e.g.*, multiplication with operators) until a stopping condition is met (convergence, unless otherwise stated), then continue fine-tuning on the target task (*e.g.*, addition/subtraction with natural language) while measuring EDL on the target task.

## 3.4. Evaluation Metrics

**EDL and normalizations** as defined in subsection 2.4.

**Performance Gap Recovery** measures how well LoRA

recovers full fine-tuning performance:

$$\text{PGR} = \frac{\text{Perf}_{\text{LoRA}} - \text{Perf}_{\text{base}}}{\text{Perf}_{\text{FullFT}} - \text{Perf}_{\text{base}}}, \tag{4}$$

where performance (Perf) is measured by accuracy or loss improvement relative to the model's zero-shot performance. PGR = 1 indicates LoRA matches full fine-tuning; PGR = 0 indicates no improvement over the base rate (zero-shot).

# 4. EDL Learning Dynamics

EDL per token scales qualitatively differently with training dataset size $n$ depending on whether the *predominant* learning process at $n$ examples is elicitation or teaching.[3] We emphasize that these signatures characterize which process dominates, not a strict dichotomy. In practice, both processes may operate simultaneously, with the scaling signature reflecting their relative contributions. In this section, we study concrete examples where we have good reasons to believe that the principal learning process is either elicitation or teaching in order to infer the appropriate dynamical behavior.

## 4.1. Elicitation: Monotonically Decreasing Returns

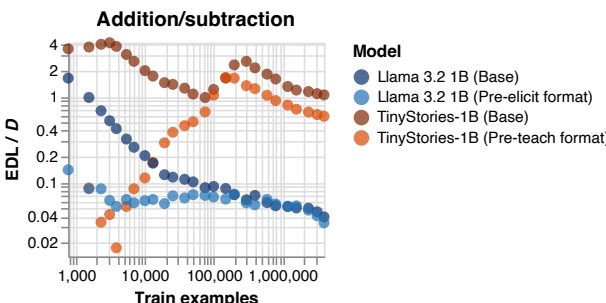

*Figure 2.* **EDL per token scaling with dataset size reveals distinct signatures for elicitation (monotonically decreasing) and teaching (increasing).** Blue points show Llama 3.2 1B, which has latent arithmetic capability; orange/brown points show TinyStories–1B, which must learn addition/subtraction from scratch. For the Llama base model (elicitation), $\text{EDL}/D$ decreases monotonically, demonstrating sharply diminishing returns that are monotonic in data. For TinyStories (teaching), $\text{EDL}/D$ increases as the model learns algorithmic structure, before eventually decreasing. Pre-elicited Llama (lighter blue) and pre-taught TinyStories variants (lighter orange shades) show how prior exposure shifts these curves: pre-*eliciting* format (Llama) removes the initial transient associated with format-learning and reduces $\text{EDL}/D$ by approximately an order of magnitude; pre-*teaching* format (TinyStories) reveals the increasing-returns phase without the initial format-learning transient. All experiments shown use LoRA rank 512.

Figure 2 shows EDL per token versus training set size for

---

[3]We propose formal definitions for elicitation and teaching as both learning processes and learning regimes in Appendix J.

Llama 3.2 1B on arithmetic tasks. Llama exhibits clear signatures of elicitation-dominated learning: EDL per token is low and decreases monotonically as dataset size increases, with each additional example providing less marginal information than the previous one. For the pre-elicited model (Llama 3.2 1B first exposed to a single example of a related arithmetic task to establish output format—a single multiplication problem when the task is addition/subtraction, and vice versa), the initial high-information regime is suppressed because format learning is already complete, and the model absorbs less information.

Decreasing EDL per token indicates that additional data is, on average, increasingly redundant with information already encoded in the parameters. This occurs when elicitation is the predominant learning process: the model surfaces capability by accessing information already learned during (pre)training.

These base models achieve 0% zero-shot accuracy because they treat prompts as text to continue rather than questions to answer (*e.g.*, completing "What is 2+2?" with "What is 3+5?"). Despite this, fine-tuning on as few as one example unlocks near-perfect performance. Notably, the corresponding instruction-tuned models (Llama 3.x 1B/3B/8B Instruct) achieve [20.2%/41.3%/56.1%] zero-shot accuracy on addition/subtraction through instruction tuning and RLHF, yet base models fine-tuned on a single example consistently outperform them (Table 10), demonstrating that low-information elicitation via SFT can unlock capability that post-training alignment procedures do not fully surface.

## 4.2. Teaching: Increasing Returns with Increasing Data

On the same task, TinyStories–1B demonstrates notably and qualitatively different scaling behavior (Figure 2). EDL per token enters a phase of *increasing returns* as data scale: additional examples increase the amount of predictive information learned per token, on average. This increasing phase is followed by a subsequent crossover to diminishing returns that onset when capability or capacity become saturated.

This non-monotonic signature reflects the dynamics of learning new capability. In the early phase, the model lacks the information needed to exploit the task structure. As more examples accumulate, the model begins learning regularity in the data. New examples make the data *more* informative on average, reducing the model's uncertainty about the emerging structure. Diminishing returns onset after the model acquires the relevant capability for the task, when the predominant learning process transitions from teaching (learning new predictive structure) to elicitation (fine-tuning existing structure).

The pre-taught variants provide additional insight into this behavior. When TinyStories–1B is first trained on randomly permuted labels (teaching task and output format but not the task algorithm), the initial decreasing phase disappears, and we instead observe increasing returns, as we isolate contributions from the model beginning to learn the algorithm without the confound of format acquisition. When first taught the algorithm using operator notation (*e.g.*, "2 + 2") before fine-tuning on natural language problems, models absorb less information throughout fine-tuning, having already acquired partial capability.

Increasing EDL per token indicates that additional data is, on average, increasingly informative for learning generalizable patterns. This occurs when teaching is the predominant learning process: the model builds new capability by encoding structural information that is missing from the parameters and must be newly acquired.

## 4.3. Crossovers Between Learning Regimes

The crossover from increasing to decreasing EDL per token marks a shift in the predominant learning process, rather than denoting a privileged threshold; both processes (elicitation and teaching) may operate simultaneously, with the EDL signature reflecting their relative contributions. Our pre-training interventions (Section 5) help disentangle these effects.

We observe crossovers in both directions, and the framework naturally accommodates mixtures of elicitation and teaching within a single training run. For Llama models on multiplication, EDL signatures are elicitation-dominated at moderate dataset sizes but cross over to teaching-dominated signatures when trained on millions of examples. Tracking performance by problem difficulty reveals the mechanism of this crossover: accuracy saturates on easy problems before medium and hard problems, with teaching signatures corresponding to improvements on harder problems that onset later in training.

In curriculum experiments on TinyStories–1B (Appendix I.3), pre-teaching multiplication (using operator notation) and then fine-tuning on progressively harder natural language problems yields elicitation signatures on easier problems, consistent with the model applying a previously-learned algorithm, but teaching signatures on harder problems that exceed the complexity and difficulty of the pre-taught examples (Figure 9). These results confirm that EDL signatures track the predominant learning process as it evolves during training.

## 4.4. Cross-Task Consistency

The same patterns hold across tasks (see Table 5 and Appendix I.1 for all task results). Nominally harder tasks, such as multiplication, show more extended increasing-returns phases for TinyStories–1B and flatter curves for weaker

elicited models. The EDL learned per token remains over an order of magnitude smaller for Llama than for TinyStories variants that must learn the algorithm from scratch, confirming that these signatures reflect fundamental differences in learning dynamics rather than task-specific artifacts.

## 5. Causal Intervention Shifts Learning Processes and Regimes

Correlation between model type and EDL signature does not establish causation; Llama and TinyStories could differ in ways unrelated to latent capability that coincidentally produce different EDL patterns. To address this, we perform causal interventions: we take a model that initially shows teaching signatures and intervene on it to show elicitation signatures by installing the relevant capability before measuring EDL. In doing this, we find that we recover the same teaching and elicitation scaling relationships, regardless of how the capability is taught (*i.e.*, pretraining vs. fine-tuning) prior to fine-tuning on the downstream task of interest.

### 5.1. Results

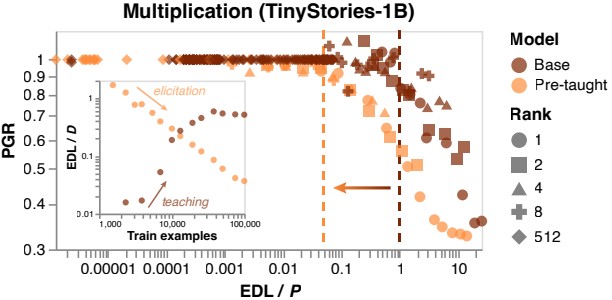

Figure 3. **Pre-teaching converts teaching tasks to elicitation tasks, shifting the capacity threshold by more than an order of magnitude.** Brown: base model (no multiplication knowledge). Light orange: pre-taught multiplication using operator notation before fine-tuning on natural language problems. The pre-taught model becomes capacity-limited at $EDL/P^* \approx 0.05$ bits/parameter, matching the elicitation regime observed for pretrained Llama models. Curves for individual LoRA ranks collapse onto each other for each model variant with consistent parameter capacity limits at model-dependent values of $EDL/P^*$ (dashed lines). Inset: $EDL/D$ scaling changes from teaching-like (increasing) to elicitation-like (decreasing) when capability is pre-taught.

Figure 3 compares scaling of PGR with EDL per parameter (inset: scaling of EDL per token with number of train set examples) for base and pre-taught TinyStories–1B models fine-tuned on multiplication. Each point compares the fraction of the total performance gap recovered by the specified LoRA rank at a given dataset size. The saturation point for each model, $EDL/P^*$, where PGR drops below 1, corresponds to the maximum learnable information density (per trainable parameter). Above this capacity limit, LoRA

begins to underperform full fine-tuning on the data.

The base model, which must learn the task algorithm without any pre-existing relevant knowledge, exhibits a capacity limit (indicated by the knee where saturation occurs) $EDL/P^* \sim 1$ bit/parameter. This corresponds to the teaching regime, where substantial information must be encoded in the model parameters to acquire the capability. In the dynamical behavior, EDL per token increases with additional train examples, as is characteristic of teaching-dominant learning. In contrast, the pre-taught model saturates the adapter capacity around 0.05 bits per parameter, a 20–fold reduction, matching the elicitation regime observed for Llama 3 models. Pre-teaching also switches the dataset-size-dependent learning dynamics from teaching-like (increasing returns from additional train examples) to elicitation-like (monotonically decreasing returns), with the pre-taught model exhibiting the same qualitative signatures as observed for the pretrained Llama models.

This intervention demonstrates that the EDL signatures of decreasing returns to information absorbed from additional data and PGR saturation at smaller information thresholds are causally related to pre-existing capability: pre-teaching converts a teaching task into an elicitation task. We observe that for the same base model, task, and fine-tuning procedure, changing purely whether the capability is latent versus absent shifts the capacity threshold by over an order of magnitude. This behavior is consistent across models and tasks (see Appendix I for all model-task combinations), supporting that prior knowledge significantly influences a model's ability to generalize from limited data.

**Out-of-distribution pre-teaching controls.** To confirm that the shift from teaching to elicitation signatures reflects knowledge of the specific algorithm rather than generic arithmetic transfer, we compare in-distribution (ID) to out-of-distribution (OOD) pre-teaching. Pre-teaching TinyStories–1B on multiplication converts multiplication fine-tuning from teaching ($EDL/P^* \approx 1.02$ bits/param, increasing returns) to elicitation ($EDL/P^* \approx 0.05$ bits/param, decreasing returns). However, pre-teaching on addition/subtraction—a related but different operation—does *not* convert multiplication to elicitation ($EDL/P^* \approx 1.56$ bits/param, increasing returns), and vice versa (Table 6). We observe that pre-teaching only the specific algorithm produces elicitation signatures, regardless of prompt format, demonstrating that EDL discriminates between genuine latent knowledge and general domain transfer. Independent validation via few-shot prompting confirms these classifications: significant improvements are observed only for models and tasks that EDL classifies as elicitation-dominated (Appendix I.9).

Additional intervention results, including curriculum learning experiments, are provided in Appendix I.

## 6. Parameter Capacity Limits

EDL per trainable parameter ($\text{EDL}/P$) provides a practical tool for predicting (1) when data will exceed model/parameter capacity and (2) when parameter-efficient methods will underperform full fine-tuning. We find that LoRA performance degrades sharply beyond a regime-dependent threshold, and this threshold can be predicted from EDL.

### 6.1. Capacity Across Model Sizes

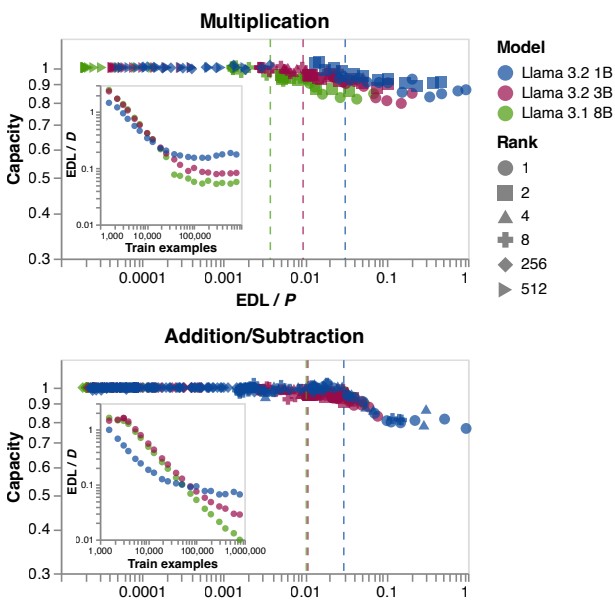

*Figure 4.* **Larger models have lower capacity thresholds for elicitation.** All models exhibit degraded compression capacity ($\text{EDL}/\text{EDL}_{\text{ref}}$) beyond a capacity threshold $\text{EDL}/P^*$ that varies with model size (dashed lines). At higher information densities exceeding $\text{EDL}/P^*$, models become capacity-limited, learning more inefficiently and less overall than full fine-tuning. Larger models require less information (fewer bits) per parameter, consistent with having more preexisting capability.

[Figure 4](#) shows capacity versus $\text{EDL}/P$ for Llama 3 models of different sizes (1B, 3B, 8B) fine-tuned on different arithmetic subsets of the DeepMind Mathematics dataset (multiplication, addition/subtraction). All model sizes extract the maximum amount of information (equivalent to full fine-tuning, $\text{EDL}/\text{EDL}_{\text{ref}} \approx 1$) when the information density $\text{EDL}/P$ is sufficiently low. Compression degrades as the threshold is exceeded and learning slows down. The saturation limit $\text{EDL}/P^*$ varies systematically with model size: larger models have lower capacity thresholds (when architecture/pretraining is fixed). This may reflect that larger models have more preexisting capability, and accordingly, they may require less information per parameter as more of the relevant representational structure already exists.

In the extreme case, Llama 3.1 8B achieves 96% accuracy on addition/subtraction after fine-tuning on a single randomly sampled example (~8 bits)—a 96 percentage point improvement from 0% zero-shot ([Table 8](#)). This demonstrates that when capabilities are latent, the information barrier to elicitation can be remarkably low.

Capacity thresholds for all models appear in [Appendix I.2](#).

### 6.2. Elicitation and Teaching Thresholds

We observe significant differences in these thresholds when comparing across regimes (Figures [3](#) and [4](#)): elicitation (Llama 3 and pre-taught TinyStories models) is associated with adapter capacity limits around 0.01–0.1 bits/parameter, whereas teaching saturates around 1 bit/parameter or more.

This two-order-of-magnitude difference has practical implications. For elicitation tasks, very low-rank adapters often suffice, as information requirements are minimal. For teaching tasks, high-rank adapters or full fine-tuning may be necessary to avoid capacity constraints, depending on dataset size and desired task performance.

These thresholds characterize where learning efficiency degrades sharply; with finite compute, models that cannot learn all regularity in the data before reaching capacity limits may absorb less total information than pre-taught models that learn efficiently throughout training. We confirm this with curriculum learning experiments: when TinyStories–1B is pre-taught on easy multiplication examples and then fine-tuned on progressively harder problems, the model learns more efficiently early in training (elicitation of the pre-taught algorithm), freeing capacity for subsequent teaching of harder problems. This results in higher final accuracy and similar total EDL as the base model trained with equivalent compute ([Appendix I.3](#)).

### 6.3. Reasoning Model Capacity

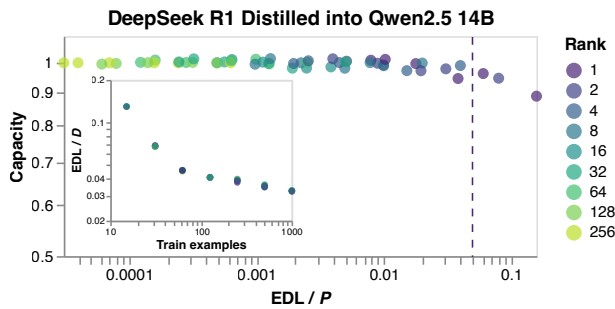

*Figure 5.* **Reasoning capability distillation shows low capacity thresholds and learning dynamics characteristic of elicitation.**

[Figure 5](#) shows capacity analysis for distillation of DeepSeek R1 into Qwen2.5 14B by fine-tuning on R1-

generated reasoning traces. The capacity limit occurs around 0.05 bits/parameter, well within the elicitation regime, with EDL per token decreasing as dataset size is scaled. Qwen2.5 1B and 32B exhibit similar elicitation-like EDL signatures: monotonically decreasing scaling trends and capacity limits of 0.04 and 0.02 bits/parameter, respectively (Tables 5 and 7). We additionally observe significant performance improvements from fine-tuning in the extreme low-information limit (Table 8): single-example fine-tuning improves multi-domain reasoning by up to 23 pp (>45% PGR for all Qwen models). Together, these findings suggest that distillation primarily improves extended, complex reasoning by surfacing latent capability rather than by teaching new reasoning ability.

### 6.4. Practical Guidance

These results provide actionable guidance for practitioners wishing to estimate $P_{min}$, the smallest adapter size with sufficient capacity to match the learning of full fine-tuning (for efficiency reasons, or to obtain an approximate upper bound on the amount of semantic information learned during training):

1. **Estimate EDL** from a pilot training run with high-rank LoRA at the target dataset size.

2. **Classify the learning process** based on EDL scaling (decreasing with dataset size = elicitation-dominated, increasing = teaching-dominated).

3. **Select an appropriate adapter size** (based on $P^*$) to ensure EDL/$P$ falls within the regime-appropriate threshold (<0.1 bit/parameter for elicitation, ~1+ bit/parameter for teaching).

## 7. Discussion

We have demonstrated that EDL serves as an effective operational metric for quantifying information absorption during training, with utility across several dimensions. Our findings are consistent across six task domains (arithmetic, chain-of-thought reasoning, reading comprehension, science QA, language modeling, and instruction following), three model families (Llama, Qwen, TinyStories), and model sizes spanning 1B to 32B parameters (Table 5).

EDL scaling dynamics characterize the predominant learning process as a function of dataset size, providing a means to distinguish elicitation-dominated from teaching-dominated learning and identify crossovers between them as capability is acquired. Elicitation is characterized by monotonically decreasing EDL per token; teaching is associated with a phase of increasing EDL per token as data scale. Evolution of this characteristic behavior provides evidence of changes in the learning process, by which elicitation be-

comes primarily teaching and vice versa. These signatures are robust across tasks and model sizes.

EDL per parameter predicts parameter capacity limits. Performance degrades sharply beyond regime-dependent thresholds: ~0.01–0.1 bits/parameter for elicitation, ~1 bit/parameter for teaching. This provides practical guidance for adapter sizing and data selection.

Finally, causal interventions confirm the assessment of learning regime and the origin of capability emergence, distinguishing elicitation from teaching and latent capability from missing capability. Pre-teaching shifts tasks from teaching-like to elicitation-like signatures, reducing capacity thresholds by more than an order of magnitude.

### 7.1. Implications for Capability Evaluation

Current evaluation practices often use fine-tuning to assess model capabilities. Our results suggest caution towards broad application of fine-tuning-based elicitation techniques in capability evaluation: fine-tuning-based evaluations may inadvertently teach capabilities rather than elicit them, especially for models without relevant pretraining exposure.

EDL signatures could inform evaluation design. If fine-tuning shows teaching signatures (increasing EDL per token phase, higher capacity thresholds), the evaluation may be measuring what the model can learn rather than what it already knows. For faithful capability assessment, evaluations should target elicitation regimes. EDL can also be used to estimate an upper bound on the amount of semantic bits learned during training, which can be used to estimate how unrealistically conservative dangerous capability evaluations are.

### 7.2. Implications for Safety

The finding that elicitation requires far less information than teaching has important safety implications. If dangerous capabilities are latent in a model, they may be surfaced with minimal data—potentially just a few examples that correctly point to the existing capability. The capacity thresholds we measure (0.01–0.1 bits/parameter for elicitation) correspond to remarkably small amounts of information. We observe instances where fine-tuning on only a single example that contains fewer than 10 bits of information can improve accuracy by over 50 percentage points (Tables 8 and 9), implying that latent capabilities may be significantly easier to access than previously assumed.

Conversely, if dangerous capabilities must be taught, the barrier is substantially higher. Understanding which capabilities are latent versus absent in deployed models is therefore both essential and practical for informed risk assessment.

Our causal interventions further demonstrate that upstream

training decisions affect downstream information requirements: pre-teaching a capability reduces the information cost of surfacing it in fine-tuning by over an order of magnitude. This implies that specific components of the training pipeline—such as particular post-training procedures, datasets, or pre-training data mixtures—may inadvertently reduce the information barrier for eliciting capabilities of concern, a consideration that EDL can help quantify.

### 7.3. Implications for Reasoning, Reinforcement Learning, and Other Post-Training Interventions

Our analysis of Qwen2.5 suggests that reasoning improvements from distillation operate in the elicitation regime. The capacity thresholds we observe (~0.02–0.05 bits/parameter) indicate that distillation surfaces latent reasoning capability rather than teaching it from scratch.

This has implications for understanding how reasoning capabilities emerge and propagate through the model training pipeline. If reasoning is primarily elicited, then models may have more latent reasoning capability than their default behavior reveals, and this capability could potentially be surfaced through other means. This also has implications for assessing whether other post-training techniques, such as compute-heavy RLHF, primarily elicit existing capability or teach new skills. Our results on Qwen and Llama demonstrate that performance improvements greater than those achieved through instruction-tuning and RLHF are accessible with few-example, information-dense supervised fine-tuning. This suggests that RLHF, while computationally expensive, may be largely elicitation.

### 7.4. Limitations

Several limitations should be noted. Our experiments focus primarily on arithmetic and reasoning tasks for which high-quality training corpora are widely available and performance can be scored against verifiable ground truth solutions; other capabilities may behave differently. We study primarily Llama and Qwen families of dense transformer models; other architectures, such as Mixture-of-Experts (MoE), may have different capacity characteristics. As EDL depends on the training algorithm, different training techniques, optimizers, learning rates, or other hyperparameters could yield different values, as well (see Appendix H.1 for additional discussion of hyperparameter sensitivity and details of all configurations tested).

We emphasize that EDL measures learning as data compression achieved through generalization, not semantic content or performance. EDL is an information-theoretic tool for assessing learning efficiency, rather than a mechanistic tool for interpreting model cognition or quantifying absolute capability.

### 7.5. Related Work

**Knowledge capacity.** Allen-Zhu & Li (2024) demonstrated that neural networks can store approximately 2 bits per parameter using synthetic datasets with precisely controlled information content. Our work extends this inquiry to practical training settings where dataset information content is difficult to estimate *a priori*. The teaching thresholds we observe (~1+ bit/parameter) align with this, whereas those consistent with elicitation are at least 1–2 orders of magnitude lower, demonstrating that far less information per parameter is needed when relevant capability already exists.

**Parameter-efficient fine-tuning.** LoRA (Hu et al., 2021) and variants enable controlled capacity constraints. Prior work (Donoway et al., 2025) showed parameter constraints proxy information constraints; we extend this with EDL-based analysis.

**Efficient capability elicitation.** Recent work shows minimal data can substantially improve reasoning: 1K examples for Qwen (Muennighoff et al., 2025), single-example RL (Wang et al., 2025), and random rewards (Shao et al., 2025). Our findings support that such gains can occur when capabilities are latent (elicitation) rather than absent (teaching).

**Information-theoretic learning.** MDL probing (Voita & Titov, 2020), information bottleneck methods (Tishby et al., 2000), and epiplexity (Finzi et al., 2026) connect compression to generalization. EDL differs from prior MDL probing in several respects: it uses population loss as a reference (measuring generalizable information rather than total compression), it accounts for multi-epoch training (standard in fine-tuning but unaddressed by single-epoch MDL), and its scaling analysis across dataset sizes reveals qualitative signatures that are not derivable from the MDL formalism. We found that repeating our analyses with MDL instead of EDL failed to reliably distinguish elicitation from teaching, whereas EDL was discriminative in all configurations tested.

### 7.6. Conclusion

EDL provides a principled, operational measure for quantifying how much predictive structure fine-tuning extracts from training data and transfers to model parameters, providing an upper bound on the amount of information (bits) required for learning. Its distinct dynamical signatures for elicitation and teaching—characterizing the predominant learning process at each dataset scale—combined with its predictive power for estimating parameter capacity limits, make it a practical tool for capability evaluation, safety assessment, and efficient training design.

## Impact Statement

This paper presents work whose goal is to advance the understanding of how language models acquire and express capabilities through training. Our findings have several potential societal implications.

**Safety evaluation.** Our framework provides tools for more accurate capability assessment. The finding that elicitation requires far less information than teaching suggests that fine-tuning-based evaluations may inadvertently teach capabilities rather than elicit them, potentially mischaracterizing deployed model risks. EDL signatures could help distinguish these cases, improving evaluation validity.

**Dual-use concerns.** Understanding that dangerous capabilities, if latent, may be surfaced with minimal data (potentially just a few examples) underscores the importance of careful capability assessment before deployment. Conversely, if dangerous capabilities must be taught, the information barrier is substantially higher and more predictable.

**Efficient training.** Our capacity threshold findings provide actionable guidance for practitioners, potentially reducing computational costs by informing appropriate adapter sizes and data requirements.

We believe the benefits of improved scientific understanding of capability emergence outweigh the risks, particularly given that our framework primarily aids evaluation rather than capability enhancement.

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

# A. Information-theoretic background and definitions

Here, we describe the information-theoretic quantities which form the foundation of our framework. Detailed explanations and full derivations for all quantities, as well as visualizations of the information-theoretic properties associated with each learning regime can be found in Donoway et al. (2026). We propose formal definitions for teaching and elicitation in Appendix J.

## A.1. A Simple Example

Let us imagine that Alice wants to efficiently communicate a fine-tuned model to Bob, who has access to both the base model and the data inputs Alice used, but not the corresponding dataset labels. Instead of sending Bob the model weights themselves, which may be very difficult for large models, Alice can alternatively send the labels of the dataset she used to train her model, which Bob can then use to train an identical copy. Alice's task is then to communicate the correct labels to Bob using the minimum number of bits necessary. The shortest codelength that can accomplish this is the minimum description length (MDL), corresponding to the minimum amount of information needed for Bob to be able to train an exact copy of Alice's model. Viewed from a different perspective, it is the minimal information required to elicit from or teach the base model a set of capabilities identical to those of Alice's model.

## A.2. Prequential MDL

Minimum description length is an uncomputable quantity; however, prequential coding provides a true upper bound on its value by using the model as a compression algorithm for the data. We employ this approximation to compute MDL in our experiments. Prequential coding trains a model on a dataset of examples provided sequentially, with the model's cross-entropy loss on each label token corresponding to the minimum amount of information required to encode and transmit that label using the model itself. The corresponding label for each example is encoded the first time it is encountered by the model, with the cross-entropy loss describing the amount of information the model requires to represent that label at the current training step. As the model trains on the data, the loss becomes smaller, and as a result, the information required for transmitting the subsequent label tokens decreases as the model improves at predicting the target distribution.

Prequential MDL is then computed by summing the model's cross-entropy loss on each label token, at the time it was encoded, over the entire dataset; this is equivalent to the total log-loss during the first epoch (Figures 6 and 7, gray

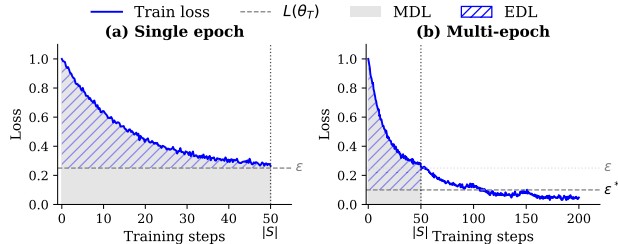

*Figure 6.* **EDL measures generalizable information extracted from finite data.** The data $S$ (size $|S|$) are fixed. (a) After one epoch, EDL (blue hatched) reflects structure learned from $S$ so far. (b) Multi-epoch training on the same data improves generalization ($L(\theta_T) = \varepsilon^* < \varepsilon$, where $T$ is the total number of training steps, $\varepsilon$ and $\varepsilon^*$ are the generalization errors after one epoch and multiple epochs, respectively, and $L(\theta_T)$ is the population loss on held out test data), yielding larger EDL. Total area under the first-epoch curve (up to $|S|$) is MDL (gray); EDL is the portion above final population loss $L(\theta_T)$. Additional training extracts more of the learnable structure in $S$ without adding new information.

shaded regions).[4] Its value corresponds to the amount of information the model needed to represent the correct labels the first time it encountered them; this is the amount of information provided to the model during training, as quantified by the model itself.

Viewing learning as compression, better training that leads to lower loss means that the model can more compactly encode and represent information about the task. Accordingly, better predictions (lower loss) carry less information—the amount of information contained in the dataset reflects how surprising that information is to the model the first time it gets seen.

## A.3. Excess Description Length

Whereas MDL describes the total amount of information contained in the training dataset according to the model, excess description length (EDL) measures the amount of information that the model ultimately extracts or absorbs about the overall distribution the training data came from. EDL quantifies the "learning cost" of generalization as the excess codelength required to encode the train data because the model had to learn to generalize.

The EDL is the predictive information the model compresses from the data over the entire learning process—the information learned from the data that generalizes beyond the train set to the actual task—with the model's generalization (achieved via training/fine-tuning) approximated by the test loss of the final trained model $L(\theta_T)$. EDL is computed as the difference between the model's total (minimum) descrip-

---

[4]Since all of the dataset labels are encoded and transmitted in a single pass over the dataset, counting losses from subsequent epochs would correspond to transmitting redundant data.

tion length of the training data (MDL) and its remaining cross-entropy on the distribution after training, visualized as the area under the first-epoch training curve above the test loss (Figures 6 and 7, hatched regions).

While it is possible for a model to overfit and perfectly predict the train set, the test loss reveals the *task-related* information that the model was unable to compress further—the residual structural information in the data that the model still cannot predict after training. This is the expected code-length of the data after the model has learned to generalize, with the EDL (comprising the remaining part of the MDL) having been "absorbed" (*i.e.*, *compressed*) into the model itself during training.

### A.4. Compression Ratio

The compression ratio, expressed as the ratio of the MDL to the data's expected residual codelength after training $(D \cdot L(\theta_T)$, where $D$ is the number of labels in the train set), describes how thoroughly a model can extract useful, generalizable information from its training data.

$$
\begin{aligned}
\text{Compression ratio} &= \frac{\text{MDL}}{D \cdot L(\theta_T)} \\
\text{Space saving} &= \frac{\text{EDL}}{\text{MDL}}
\end{aligned}
\tag{5}
$$

A high compression ratio indicates that a model can compress most of the information it initially required to encode the training data by learning the general, underlying features of the task, rather than by memorizing specific examples. In contrast, a low compression ratio implies that the model was unable to compress most of the information in the training data, either because it was unable to learn the generalizable features or because there are few predictive patterns left.

## B. Task Details

**Arithmetic** tasks use subsets of the DeepMind Mathematics dataset, which can also be procedurally generated. We study addition/subtraction and multiplication separately, as these differ in difficulty and provide natural intervention targets. Problems are presented in natural language (*e.g.*, "What is the sum of 23 and 45?") with numerical answers. Dataset sizes range from 1 example to 4 million examples.

**Reasoning** tasks use chain-of-thought mathematical, scientific, and technical problem solving, where we score the full reasoning trace plus answer. This allows measurement of how much information is required to elicit extended reasoning behavior.

**Language modeling** tasks use the TinyStories-v2 dataset (Eldan & Li, 2023), composed of over 2.6 million simple English short stories with no technical content. As a basic

language modeling task, this enables assessment of teaching from scratch when no information is known (randomly initialized models), as well as comparison to elicitation (pre-trained models, which already have complex language capabilities).

**Reading comprehension** tasks use the BoolQ dataset, which requires interpreting self-contained short passage-question pairs and determining whether the answer to the question is true or false based on the content of the passage only.

**Instruction following** tasks use the Alpaca dataset (Taori et al., 2023), which consists of 52K instruction-response pairs spanning diverse tasks including open-ended generation, summarization, classification, and question answering. This dataset tests whether base models can be elicited to follow natural language instructions, a capability that is typically surfaced through instruction tuning and RLHF. We use Alpaca to evaluate EDL signatures in an open-ended, multi-task setting that contrasts with the structured, single-domain arithmetic tasks.

**Science question answering** tasks use the ARC dataset (Clark et al., 2018), which consists of multiple-choice science questions drawn from standardized tests. ARC–Easy contains questions that are answerable by simple retrieval or co-occurrence methods, while ARC–Challenge contains questions that require more complex reasoning and are not solvable by simple baselines. Both subsets test whether models can be elicited to apply scientific knowledge to novel questions. For evaluation, we use balanced accuracy to account for class imbalances in the answer choice distribution and employ logit bias correction to determine unbiased zero-shot baselines (Appendix D).

## C. Model Details

**Llama 3.2 1B, 3B and Llama 3.1 8B** are pretrained on large, diverse corpora including mathematical content. These models have latent arithmetic, reasoning, and instruction-following capabilities that can be elicited with fine-tuning. They serve as our primary "elicitation" condition. We also compare a randomly initialized variant of Llama 3.2 1B, which serves as a "teaching" condition where all capability must be learned from scratch, absent any pre-existing meaningful representations.

**TinyStories–1B** uses the Llama 3.2 1B architecture but is pretrained exclusively on the TinyStories-v2 corpus (Eldan & Li, 2023) (simple English short stories for children which use only a limited vocabulary, containing no numerical digits, mathematical operators, or technical content). This model must learn arithmetic from scratch during fine-tuning, providing a clean "teaching" condition.

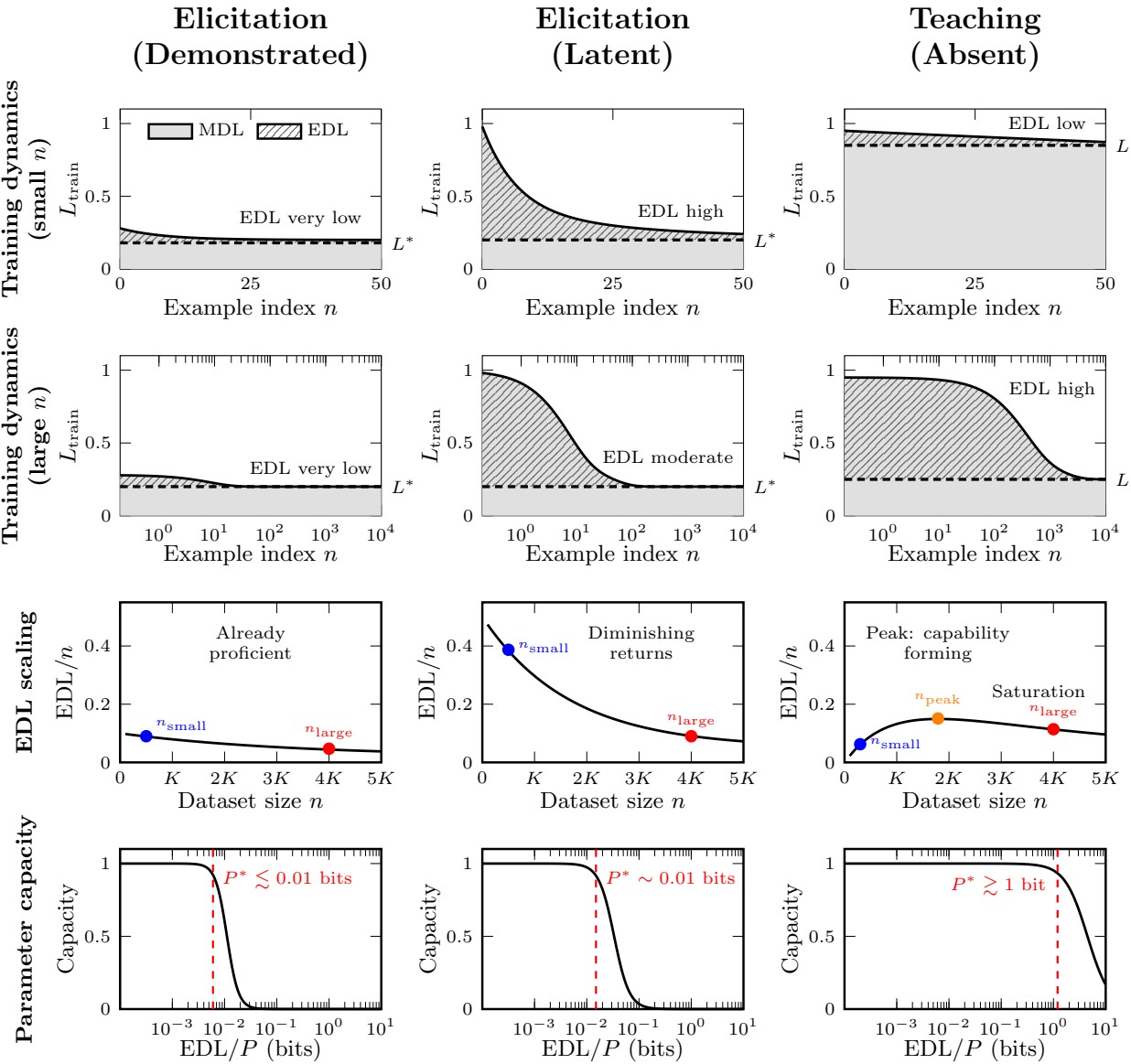

*Figure 7.* **Conceptual diagram illustrating how EDL robustly distinguishes elicitation of existing capability (left and center columns) from teaching (right column), even when zero-shot performance cannot.** Models with latent capabilities may behave the same as models that lack capability (similar initial loss), but small amounts of information suffice to elicit relevant pretrained knowledge if it exists. Conceptual diagrams of eliciting demonstrated capabilities (left column), eliciting latent capabilities (center column), and teaching new capabilities (right column) in small and large data regimes. (*Top row*: small $n$, second row: large $n$) The minimum description length (MDL) is the total area under the training curve $L_{\text{train}}$ (all gray shaded areas), representing the number of bits provided to the model (through the labels of the fine-tuning dataset), as quantified by the model itself. The final converged test loss (after training for as many epochs as necessary) $L^*$ represents the model's asymptotic error on the true data distribution when trained on $n$ unique examples. The excess description length (EDL) is the area between the training curve $L_{\text{train}}$ and the generalization error $L^*$, which represents the generalizable information that is absorbed by the model's parameters during training, corresponding to the reduction in its prediction error on the underlying data distribution. (*Third row*) Scaling behavior of EDL per example (EDL/$n$ as dataset size $n$ increases, where dataset size is normalized by the total number of constituent concepts $K$ that must be known for task mastery). Models that already know the necessary task components but improve their ability to employ that knowledge (elicitation) absorb less generalizable information with additional examples, whereas models that must learn the requisite concepts from scratch experience increasing returns when developing fundamental task capability. (*Bottom row*) Capacity (fraction of information absorbed compared to full fine-tuning; EDL/EDL$_{\text{ref}}$) versus EDL per parameter reveals different parameter capacity limits for elicitation and teaching: models which require less information to achieve maximal task performance (elicitation) saturate parameter capacity at lower information thresholds than models which require more information to learn a task.

**Qwen2.5 1.5B, 14B, and 32B** are pretrained on large corpora primarily composed of examples which provide a foundation for common sense, expert technical knowledge, and reasoning capabilities. We use these models to study reasoning capability elicitation, particularly at larger scales (14B, 32B). We compare to the corresponding reasoning-focused models (Qwen2.5 1.5B, 14B DeepSeek-R1-Distill) created by distilling DeepSeek R1 into the respective Qwen2.5 base models.

## D. Methodology

We elicit pretrained-only base models by fine-tuning on several popular benchmarks and datasets, which include multiple choice and natural language generation tasks (Deepmind Mathematics (Saxton et al., 2019), BoolQ (Clark et al., 2019), ARC–Easy/Challenge (Clark et al., 2018), TinyStories-v2 (Eldan & Russinovich, 2023), R1-generated reasoning traces filtered for accuracy using prompts from the s1K dataset (Muennighoff et al., 2025)), and Alpaca (Taori et al., 2023). We regard a model as capable of being elicited on a particular task if multi-shot prompting or removing biases in the model's logit distribution (for multiple choice tasks) results in significantly improved performance relative to the model's original zero-shot baseline.

We enforce information constraints on the models and training process through two methods: restricting the number of trainable parameters and restricting the size of the dataset used for training. We refer to these settings as "parameter-controlled" and "dataset-controlled," respectively. We fine-tune on the full dataset and use LoRA (Hu et al., 2021) for all parameter-controlled experiments, as we find it to yield the best performance per parameter; for the data-controlled setting, we truncate the training dataset at varying fractions of the total dataset size and perform full-model fine-tuning. In both settings, we train to convergence in all experiments.

To distinguish between teaching and elicitation, we perform comparative experiments on pretrained and randomly initialized models which share the same architecture, applying the same training procedures (supervised fine-tuning) on identical datasets. Because the randomly initialized models contain no preexisting learned representations, improvements on the tasks indicate teaching entirely new capabilities from scratch.

To measure teaching and elicitation in models with a pre-existing knowledge base, we first pretrain a randomly initialized Llama 3.2 1B model on a corpus of simple English-language short stories which use only a small vocabulary (TinyStories-v2) to teach isolated language skills without introducing additional capabilities, such as arithmetic or science proficiency. We then post-train these narrow, "language-only" models on the same tasks (mentioned pre-

viously) with the same fine-tuning procedures as the corresponding pretrained base models, comparing performance with scaling of MDL and trainable parameters.

Additional details about each of these settings, as well as all training configurations and hyperparameters can be found in Appendix G.

## E. Pre-training Intervention Details

For all experiments in this paper, all pre-training interventions are performed using full fine-tuning (as opposed to LoRA).

### E.1. Format Pre-training

#### E.1.1. LLAMA 3 MODELS

For all Llama 3 base models, we perform format pre-training to elicit correctly formatted responses by fine-tuning on out-of-distribution examples which share the same output formatting. This is because training with random labels can alter the model's inductive bias for the output distribution beyond simply aligning the outputs to the target task's formatting requirements, since models may learn (potentially spurious) features of the pre-elicitation distribution that may interact or conflict with existing knowledge.

For example, pre-eliciting for a binary choice task using an example with a randomized binary label from the target distribution may bias the model towards responding with the opposite label (if the randomized label is incorrect and opposite to a high probability prediction). Alternatively, randomized labels that happen to be correct may bias the model towards the correct distribution but remain unaccounted for in the EDL, which is computed on the target distribution.

For Llama 3.2 1B (pre-elicit arithmetic), we fine-tune to convergence[5] on a single arithmetic problem sourced from a different domain and prompt format than the target task. For example, if the target task is addition/subtraction problems expressed in natural language (*e.g.*, "What is the sum of 2 and 1?"), we first "pre-elicit" by fine-tuning on a single example of a multiplication problem expressed in operator notation (*e.g.*, "$3 \times 4$"). This establishes the output format without providing task-specific information.

We obtain similar results using LoRA vs. full fine-tuning for the pre-elicitation fine-tuning step. The experiments in this paper all use full fine-tuning for any pre-elicitation step(s).

---

[5]"Convergence" is determined by maximal performance on the chosen validation metric (either maximization of the validation accuracy or minimization of the validation loss); for a single example, this usually occurs after a single training step.

E.1.2. TINYSTORIES–1B

For TinyStories–1B (Pre-teach format) on arithmetic tasks, we pre-train by fine-tuning on examples with randomly permuted labels until convergence (formatting of `prompt` vs. **`label`** respectively denote masked and scored components for loss computation):

- Multiplication (mult.) example (`prompt` and **`label`**):
  ```
  Question:
  2 * 3
  Answer:
  7
  ```

- Addition/subtraction (add/sub):
  ```
  Question:
  2 + 3
  Answer:
  4
  ```

This teaches:

- Numerical digits (0-9), which TinyStories–1B has never seen

- Output format (respond with only the numerical answer)

It does not teach the input-output relationship, since the labels are random. Since TinyStories–1B does not have a meaningful inductive bias for the task (unlike models which have encountered similar material during diverse pretraining), pre-teaching using randomized labels also does not affect the output distribution beyond teaching formatting.

For the experiments in this paper, the pre-training examples used for teaching formatting are sourced from the same arithmetic domain as the target task but have different prompt formatting. We observe similar results regardless of the pre-training domain or prompt (input) formatting used, as long as the output format is the same as the target task (e.g., output the answer as a single number).

### E.2. Algorithm Pre-teaching

For TinyStories–1B (Pre-teach mult.) and (Pre-teach add/sub), we fine-tune on problems expressed with operators (formatting of `prompt` vs. **`label`** respectively denote masked and scored components for loss computation):

- Multiplication (mult.) example (`prompt` and **`label`**):
  ```
  Question:
  2 * 3
  Answer:
  6
  ```

- Addition/subtraction (add/sub):
  ```
  Question:
  2 + 3
  Answer:
  5
  ```

We perform full fine-tuning for a single epoch on 4 million unique examples. The final model is then used for the corresponding natural language task.

## F. Token Accounting and Output Scoring

For arithmetic tasks, we score only the numerical answer tokens. The prompt (e.g., "What is the sum of 23 and 45?") and any formatting tokens are excluded from both MDL computation and test loss evaluation.

For reasoning tasks (Qwen), we score the full response including chain-of-thought reasoning final answer. This captures the information required to elicit the full reasoning behavior.

All losses are computed in nats (natural logarithm base). For EDL, MDL, and all other information measures, we convert to bits by dividing by $\ln 2$ (conversion: bits = nats / $\ln 2 \approx$ nats$/0.693$).

## G. Training Details

### G.1. Parameter-Controlled Training

To measure the capacity of post-training procedures to elicit versus teach models relevant capabilities, we train and evaluate models in various settings in which their performance improvements can be attributed to either the application of relevant preexisting abilities or learning of new capabilities.

We elicit pretrained-only Llama 3 base models (Llama-3.2-1B, Llama-3.2-3B, Llama-3.1-8B) by fine-tuning on several popular benchmarks and datasets, which include multiple choice and natural language generation tasks (BoolQ, ARC-Easy/ARC-Challenge, TinyStories-v2, DeepMind Mathematics, Alpaca). These models have been trained on large, diverse corpora which approximately encompass the entire internet, imbuing them with knowledge of task-relevant concepts. We consider a model as capable of being elicited on a particular task if we observe a significant difference between its multi-shot and zero-shot performance on that task. Additionally, for multiple choice tasks, we employ logit bias correction for distinguishing elicitation from teaching, in which we subtract the model's prior(s) on the dataset's answer choice distribution to determine its "unbiased" zero-shot performance on the task, using this as one baseline for the minimum performance that can be elicited from the model.

We enforce information constraints on the models and training process through two methods: restricting the number of trainable parameters and restricting the size of the dataset used for training. We refer to these settings as "parameter-controlled" and "dataset-controlled," respectively. Both methods effectively constrain the amount of information that can be added to the model during training; the former places strict architectural constraints on the amount of (new) information that can be stored in the model as well as the amount by which its original representations can maximally change, and the latter provides a bound defined by the information content of the dataset.

In the parameter-controlled setting, we use low-rank adaptation as a parameter efficient fine-tuning (PEFT) technique to restrict the number of trainable parameters while still aiming to retain as much performance per parameter as possible. We use LoRA (Hu et al., 2021) as for all parameter-controlled experiments, as we find it to yield the best performance per parameter (additional details about parameter-controlled training can be found in Appendix G.1). For the data-controlled setting, we truncate the training dataset at varying fractions of the total dataset size and use the same hyperparameters for each (model, dataset) configuration, ensuring that the same examples are seen in the same order by each model during the first epoch so that its training dynamics up to each successive example (batch) are the same, irrespective of dataset size. All training configurations and hyperparameters can be found in Appendix H.1.

To distinguish between teaching and elicitation, we perform comparative experiments on pretrained and randomly initialized models which share the same architecture, applying the same training procedures (supervised fine-tuning) on identical datasets. Because the randomly initialized models contain no preexisting learned representations, any and all improvements on the tasks can be directly and unambiguously attributed to teaching entirely new capabilities from scratch.

To measure teaching and elicitation in models with a preexisting knowledge base, we first pretrain a randomly initialized Llama-3.2-1B model on a corpus of simple English-language short stories which use only a small vocabulary (TinyStories-v2) to teach isolated language skills without introducing additional capabilities, such as arithmetic or science proficiency. We then post-train these narrow, "language-only" models on the same tasks (mentioned previously) with the same fine-tuning procedures as the corresponding pretrained base models, comparing performance with scaling of MDL and trainable parameters.

### G.1.1. LoRA AND RANDOM SPARSE LoRA TRAINING

LoRA is applied to all projections in all transformer layers ($Q$, $K$, $V$, $O$, $G$, $U$, $D$). For all experiments, we use the standard parameterization used in the Hugging Face `peft` library (Mangrulkar et al., 2022): Kaiming initialization for $A$ with scale $1/\sqrt{d_{\text{in}}}$, zero initialization for $B$, the same learning rate for both $A$ and $B$, and $\alpha = 32$. All experiments use the standard scaling $\alpha/2r$, where $r$ is the LoRA rank. Dropout is set to 0.

### G.1.2. OTHER PEFT METHODS

We also evaluated other LoRA variants, such as DoRA and PiSSA, for their parameter efficiency, finding that LoRA was most parameter efficient for our settings and tasks.

Other parameter-efficient fine tuning techniques, such as soft token methods (including prefix tuning, P-tuning, and prompt tuning), were also evaluated for their efficiency in improving capabilities through elicitation and teaching. We find these to be less parameter efficient than LoRA. This is because there is a lower bound on the number of tunable parameters that can be used, determined by the hidden dimension of the model. For the models tested, the hidden dimension $d = \{2048, 3172, 4096\}$ for Llama 3.2 1B, Llama 3.2 3B, and Llama 3.1 8B, respectively, resulting in performance worse than tuning over an order of magnitude fewer LoRA parameters.

### G.2. Data-Controlled Training

### G.3. "Language-only" Pretraining on TinyStories-v2

Because the TinyStories pretraining corpus contains no specialized knowledge and purely teaches basic, fundamental language skills, this setup provides a testbed for assessing and measuring the information required for learning entirely new capabilities.

## H. Hyperparameter Configurations, Additional Training Details, and Computational Requirements

### H.1. Hyperparameter Details

This section provides complete hyperparameter specifications for all experiments reported in this paper.

### H.1.1. COMMON TRAINING CONFIGURATIONS

See Table 1 for training hyperparameters used for all experiments and results reported in this paper.

Table 2 describes the configurations used for all reported LoRA experiments and results.

### H.1.2. HYPERPARAMETER SENSITIVITY

We additionally tested several other optimizers (SGD, Adafactor, AdamW), learning rates and learning rate sched-

*Table 1.* Common training configuration across all experiments.

| Parameter | Value |
|---|---|
| Optimizer | AdamW |
| $\beta_1$ | 0.9 |
| $\beta_2$ | 0.999 |
| Weight decay | 0.01 |
| LR scheduler | Constant |
| Gradient clipping | 1.0 |
| Precision | bfloat16 |
| Stopping criterion | Validation loss convergence |
| Seeds per config | 3 |

*Table 2.* LoRA configuration for all experiments.

| Parameter | Value |
|---|---|
| Target modules | Q, K, V, O, G, U, D (all layers) |
| Rank sweep | 1, 2, 4, 8, 16, 32, 64, 128, 256, 512 |
| $\alpha$ | 32 |
| Scaling | $\alpha/2r$ |
| Dropout | 0 |
| A initialization | Kaiming (scale $1/\sqrt{d_{\text{in}}}$) |
| B initialization | Zero |

ules (linear, cosine, and custom schedules designed to match LoRA training dynamics to full fine-tuning; with and without warmup), and LoRA variants (vanilla LoRA, RSLoRA, DoRA, PiSSA). We deliberately report AdamW + constant learning rate because this configuration is the most principled and straightforward choice for EDL measurement, as alternative configurations introduce confounds that could produce misleading results if presented without careful renormalization (removing scale-dependent differences that are not physically meaningful):

1. AdamW consistently yields the best-performing models, fastest convergence, and EDL values closest to the supremal EDL (the algorithm-independent quantity, corresponding to the optimal learning algorithm in the hypothesis class). Using the best optimizer provides the tightest empirical bound for estimating the minimal information cost of elicitation vs. teaching. Reporting a suboptimal optimizer would overestimate the information required.

2. Decaying LR schedules (cosine, linear) make the effective learning rate dependent on dataset size, since models trained on fewer examples experience faster decay per example. This artificially suppresses learning on smaller datasets (where decay occurs more quickly) and inflates apparent EDL differences across scales that reflect the schedule rather than the information content. Using constant learning rate ensures that such

confounds are not introduced by scheduler artifacts, enabling fair and straightforward comparison of information absorbed across scales.

3. RSLoRA rescales updates by $1/\sqrt{r}$, making the effective learning rate rank-dependent. Comparing capacity across ranks then conflates optimization dynamics with information saturation, requiring renormalization to draw valid conclusions.

AdamW with constant learning rate avoids these confounds while yielding EDL close to the supremal (algorithm-independent) value. After appropriate renormalization to remove scale-dependent artifacts, all tested configurations yield identical qualitative signatures and same-order-of-magnitude capacity limits for runs that converge to similar test losses. As renormalization is not straightforward, we omit those results to avoid misinterpretation and introduction of subtleties that confound capacity estimation.

Importantly, these hyperparameter choices primarily affect the low-data, low-parameter regime of single-epoch training, where differences in learning dynamics lead to different test losses and therefore different EDL. Larger datasets and multi-epoch training both smooth out these differences as models converge to similar minima, amortizing unfavorable learning dynamics and optimizer-dependent transients over more training steps.

This is a primary advantage of EDL over single-epoch MDL and related quantities: EDL provides a straightforward and interpretable measure of information absorbed in settings where multi-epoch training improves generalization, which are common in practice. If a small dataset is sufficient to elicit latent capability, multi-epoch training should find the minimum regardless of hyperparameters, and EDL will be small relative to the parameter count.

### H.1.3. DEEPMIND MATHEMATICS EXPERIMENTS

See Table 3.

### H.1.4. QWEN REASONING EXPERIMENTS

See Table 4.

### H.1.5. OTHER TASKS

All other experiments use learning rate 1e-4 for LoRA and 2e-5 for full fine-tuning. Batch sizes for Alpaca is 32 for Llama 3.2 1B, 16 for Llama 3.2 3B, and 8 for Llama 3.1 8B. For BoolQ and ARC-Easy/Challenge, batch size is 128 for Llama 3.2 1B, 32 for Llama 3.2 3B, and 16 for Llama 3.1 8B. For TinyStories-v2, batch size is 8 for both the pre-trained and randomly initialized models.

*Table 3.* Hyperparameters for DeepMind Mathematics experiments (arithmetic tasks).

| Model | Method | Learning Rate | Batch Size | Eff. Batch Size | GPUs |
|---|---|---|---|---|---|
| TinyStories–1B | LoRA
Full FT | $3.53 \times 10^{-4}$
$2 \times 10^{-5}$ | 128 | 1024 | 8×H100 |
| Llama 3.2 1B | LoRA
Full FT | $3.53 \times 10^{-4}$
$2 \times 10^{-5}$ | 128 | 1024 | 8×H100 |
| Llama 3.2 3B | LoRA
Full FT | $1 \times 10^{-4}$
$2 \times 10^{-6}$ | 128 | 1024 | 8×H100 |
| Llama 3.1 8B | LoRA
Full FT | $1 \times 10^{-4}$
$2 \times 10^{-6}$ | 128 | 1024 | 8×H100 |

*Table 4.* Hyperparameters for Qwen reasoning distillation experiments.

| Model | Method | Learning Rate | Batch Size | Eff. Batch Size | GPUs |
|---|---|---|---|---|---|
| Qwen2.5 1.5B | LoRA | $1 \times 10^{-4}$ | 1 | 8 | 8×H100 |
| Qwen2.5 14B | LoRA | $1 \times 10^{-4}$ | 1 | 8 | 8×H100 |
| Qwen2.5 32B | LoRA | $1 \times 10^{-4}$ | 1 | 8 | 8×H100 |

## H.2. Computational requirements

All experiments were performed using a cluster of 8x H100s.

## I. Additional Results

This section presents additional experimental results that support the main text findings. We organize results by experiment type to demonstrate consistency across models, tasks, and configurations.

### I.1. EDL Scaling Signatures Across Models & Tasks

Table 5 describes EDL per token vs. training set size for model-task combinations. The same qualitative patterns hold:

- Llama models (all sizes) exhibit monotonically decreasing EDL/token on all tasks, consistent with elicitation-dominated learning.

- Qwen models (all sizes) exhibit monotonically decreasing EDL/token on reasoning tasks, consistent with elicitation-dominated learning.

- TinyStories–1B exhibits non-monotonic (increasing then decreasing) EDL/token on arithmetic tasks, consistent with teaching-dominated learning transitioning to elicitation.

- The randomly initialized variant of the Llama 3.2 1B architecture exhibits non-monotonic EDL/token signatures, consistent with teaching-dominated learning (as is expected for a model that entirely lacks meaningful

representational structure), followed by elicitation once basic language modeling capability has been acquired.

- Pre-training interventions shift signatures as predicted (teaching-like → elicitation-like).

### I.2. Parameter Capacity Curves Across Models

Table 6 and Table 7 provide capacity limits (EDL/$P^*$) for all model-task combinations with datasets that are large enough to exceed parameter capacity (larger than ~10K examples, as well as Qwen-R1/s1K Reasoning). The capacity thresholds vary systematically:

- **Elicitation regime:** EDL/$P^* \approx$ 0.01–0.1 bits/parameter

- **Teaching regime:** EDL/$P^* \approx$ 1+ bit/parameter

- Larger models have lower elicitation thresholds

#### I.2.1. CAUSAL INTERVENTION

**Intervention Design.** We pre-train TinyStories–1B on multiplication problems expressed with operators (*e.g.*, "2 * 3 = 6") until the model achieves strong performance. This creates TinyStories–1B (Pre-teach mult.), which now possesses multiplication capability. We then measure EDL when fine-tuning this models on multiplication problems expressed in natural language (*e.g.*, "What is the product of 3 and 4?").

We perform the same intervention procedure for addition/subtraction, pre-training on addition and subtrac-

*Table 5.* Summary of EDL scaling signatures across model-task combinations. ↓: monotonically decreasing (elicitation-dominated). ↑↓: non-monotonic with initial increase (teaching-dominated, then elicitation). Peak $n$: approximate dataset size at which EDL/token peaks (for teaching signatures). Dashed entries indicate that the peak $n$ occurs at the start of training (initial example/batch) and diminishing returns onset (nearly) immediately.

| Task | Model | Signature | Peak $n$ | Notes |
|---|---|---|---|---|
| Addition/Subtraction | Llama 1B–8B | ↓ | – | Latent capability |
| | Llama 1B–8B (pre-elicit) | ↓ | – | Format learned |
| | TinyStories–1B (base) | ↑↓ | $\sim$300K | Must learn algorithm |
| | TinyStories–1B (pre-teach format) | ↑↓ | $\sim$150K | Isolates algorithm learning |
| | TinyStories–1B (pre-teach add/sub) | ↓ | – | Converts to elicitation |
| Multiplication | Llama 1B–8B | ↓ | – | Latent capability |
| | Llama 1B–8B (pre-elicit) | ↓ | – | Format learned |
| | TinyStories–1B (base) | ↑↓ | >4M | Must learn algorithm |
| | TinyStories–1B (pre-teach format) | ↑↓ | $\sim$4M | Isolates algorithm learning |
| | TinyStories–1B (pre-teach mult.) | ↓ | – | Converts to elicitation |
| Reasoning | Qwen 1.5B–32B | ↓ | – | Latent capability |
| TinyStories-v2 | Llama 1B (pretrained base) | ↓ | – | Already proficient |
| | Llama 1B (random initialization) | ↑↓ | $\sim$1K | Must learn language |
| BoolQ | Llama 1B–8B (pretrained) | ↓ | – | Latent capability |
| | TinyStories–1B (base) | ↓ | – | Format acquisition |
| ARC–Easy | Llama 1B–8B (pretrained) | ↓ | – | Latent capability |
| | TinyStories–1B (base) | ↓ | – | Format acquisition |
| ARC–Challenge | Llama 1B–8B (pretrained) | ↓ | – | Latent capability |
| | TinyStories–1B (base) | ↓ | – | Format acquisition |
| Alpaca | Llama 1B–8B | ↓ | – | Latent capability |

tion problems expressed in operator notation to create TinyStories–1B (Pre-teach add/sub).

Results from causal intervention experiments can be found in tables 5, 6, 8 and 9.

### I.3. Curriculum Learning and Crossover Between Regimes

The main text demonstrates that EDL signatures distinguish elicitation from teaching in controlled settings where the predominant learning process is distinctly one or the other. Here, we investigate settings where *both* processes operate within a single training run, examining how EDL signatures evolve as the learning process shifts.

#### I.3.1. LLAMA: CROSSOVER TO TEACHING AT LARGE DATASET SIZES

When Llama 3.2 1B/3B and Llama 3.1 8B are fine-tuned on multiplication with moderate dataset sizes ($n \lesssim$ 100K examples), EDL per token decreases monotonically, consistent with elicitation of latent arithmetic capability. However, when trained on hundreds of thousands to millions of examples, EDL signatures cross over to teaching-dominated behavior: the slope of EDL per token changes from negative to positive, indicating that additional examples provide increasing marginal information as the model begins learning

beyond its pre-trained capability.

Figure 8 tracks performance by problem difficulty level throughout training. Accuracy saturates on easy problems (those with fewer digits) before medium and hard problems. The onset of teaching signatures corresponds to improvements on harder problems that the model's pre-trained knowledge is insufficient to solve, confirming that EDL correctly identifies the transition from surfacing existing capability to acquiring new capability.

#### I.3.2. CURRICULUM LEARNING ON TINYSTORIES-1B

To study the transition between regimes under controlled conditions, we pre-teach TinyStories-1B on easy multiplication examples (problems with operations that require manipulating two or fewer digits) using operator notation until convergence. We then fine-tune on natural language multiplication problems presented in a curriculum of increasing difficulty: single-digit → two-digit → three-digit → ... → up to 12-digit operations.

**Results.** On earlier, easier problems in the curriculum, the model exhibits elicitation signatures (Figure 9): EDL per token decreases monotonically, consistent with the model applying the previously learned multiplication algorithm to problems within the complexity range it was pre-taught. On later, harder problems that exceed the complexity of

*Table 6.* Parameter capacity thresholds (EDL/$P^*$) on DeepMind Mathematics tasks. EDL/$P^*_{\text{EDL}}$ is the threshold where EDL/EDL$_{\text{ref}}$ drops below 0.95; EDL/$P^*_{\text{PGR}}$ is the threshold where PGR drops below 0.95. The close correspondence between these thresholds validates that EDL-based capacity limits predict performance-based capacity limits. All values in bits/parameter. Dashes indicate experiments not run. Asterisk (*) indicates that PGR $\geq$ 0.95 for all dataset sizes tested.

| | | Addition/Subtraction | | Multiplication | |
|---|---|---|---|---|---|
| **Category** | **Model** | EDL/$P^*_{\text{EDL}}$ | EDL/$P^*_{\text{PGR}}$ | EDL/$P^*_{\text{EDL}}$ | EDL/$P^*_{\text{PGR}}$ |
| Llama 3 (Elicitation) | Llama 3.2 1B | 0.03 | * | 0.03 | 0.14 |
| | Llama 3.2 3B | 0.01 | * | 0.009 | * |
| | Llama 3.1 8B | 0.01 | * | 0.004 | * |
| Pre-elicited Llama 3.2 1B | Pre-elicit mult. | * | * | 0.07 | 0.06 |
| | Pre-elicit add/sub | * | * | 0.07 | 0.06 |
| TinyStories 1B (Teaching) | TinyStories 1B (Base) | 2.21 | 2.23 | 0.70 | 1.02 |
| Pre-taught TinyStories 1B | Format only | 1.04 | 0.96 | 0.82 | 0.73 |
| | Pre-teach mult. | 0.93 | 1.07 | 0.06 | 0.05 |
| | Pre-teach add/sub | 1.81 | 1.50 | 1.48 | 1.56 |

*Table 7.* Parameter capacity thresholds (EDL/$P^*$ at which EDL/EDL$_{\text{ref}}$ = 0.95) on TinyStories-v2, Alpaca, and Qwen-R1/s1K tasks. All values in bits/parameter.

| **Task** | **Family** | **Model** | EDL/$P^*$ |
|---|---|---|---|
| TinyStories-v2 | Llama 3.2 1B | Base | 0.02 |
| | | Random init. | 1.2 |
| Alpaca | Llama 3.2 | 1B | 0.03 |
| | | 3B | 0.02 |
| | Llama 3.1 | 8B | 0.02 |
| Qwen–R1/s1K | Qwen 2.5 | 1.5B | 0.04 |
| | | 14B | 0.05 |
| | | 32B | 0.02 |

the pre-taught examples, the model transitions to teaching signatures: EDL per token increases with dataset size and capacity thresholds rise (EDL/$P^* \approx 0.79$ bits/parameter), reflecting the need to acquire new structural knowledge to solve problems beyond the model's pre-existing capability.

**Comparison to base model.** Under equivalent compute budgets, the pre-taught model achieves both higher final accuracy and similar total EDL than the base TinyStories-1B model trained from scratch on the same curriculum. This is not contradictory: the base model's learning stalls before discovering efficient algorithms, causing it to memorize inefficiently and hit capacity limits at lower accuracy. The pre-taught model, having already internalized the core algorithm, learns efficiently early in training (elicitation), freeing parameter capacity for subsequent learning of harder problems (teaching). This confirms the prediction from Appendix J.4 that pre-teaching enables more efficient use of finite parameter capacity.

**Implications.** These curriculum experiments demonstrate three important properties of EDL:

1. EDL signatures track the predominant learning process as it evolves within a single training run, accommodating mixtures and transitions between elicitation and teaching.

2. The transition from elicitation to teaching corresponds to meaningful changes in what the model is learning (easy $\rightarrow$ hard problems), not artifacts of dataset size or training dynamics.

3. Curriculum design can improve learning efficiency by enabling models to elicit pre-existing knowledge before teaching new knowledge, rather than attempting to learn everything simultaneously. This is reflected in EDL as more efficient early learning (lower EDL/token on easy problems) that frees capacity for later teaching (higher total EDL on hard problems).

### I.4. BoolQ Results

We additionally validate our findings on BoolQ, a reading comprehension task:

- Data constraints and parameter constraints yield equivalent EDL at matched performance levels, confirming that EDL captures the relevant information regardless of how constraints are imposed.

- Llama models exhibit elicitation-like signatures (low EDL/token, diminishing returns).

- TinyStories–1B, which has basic language capability, shows elicitation-like signatures for the question-answering format.

### I.5. Performance Gap Recovery Details

Table 8 and Table 9 summarize performance gap recovery (PGR) across model-task combinations for extreme low-information settings.

*Table 8.* Performance gap recovery with minimal information. PGR measures the fraction of full fine-tuning performance achieved. Numbers in parentheses indicate absolute accuracy improvement (percentage points).

| Architecture | Model | Task | 1 Example | 10 Params | 100 Params |
|---|---|---|---|---|---|
| Llama 3 | Llama 3.2 1B | Addition/Subtraction | 0.66 (+66 pp) | 0.65 (+65 pp) | 0.71 (+71 pp) |
| | | Multiplication | 0.31 (+31 pp) | 0.30 (+30 pp) | 0.31 (+31 pp) |
| | Llama 3.2 3B | Addition/Subtraction | 0.86 (+86 pp) | 0.87 (+87 pp) | 0.89 (+89 pp) |
| | | Multiplication | 0.54 (+54 pp) | 0.65 (+65 pp) | 0.68 (+68 pp) |
| | Llama 3.1 8B | Addition/Subtraction | 0.96 (+96 pp) | 0.91 (+91 pp) | 0.96 (+96 pp) |
| | | Multiplication | 0.67 (+67 pp) | 0.76 (+76 pp) | 0.81 (+81 pp) |
| Llama 3 | TinyStories–1B | Addition/Subtraction | 0.0 (+0 pp) | 0.0 (+0 pp) | 0.0 (+0 pp) |
| | TinyStories–1B | Multiplication | 0.0 (+0 pp) | 0.0 (+0 pp) | 0.0 (+0 pp) |
| Qwen2.5 | Qwen2.5 1.5B | Reasoning | 0.53 (+7 pp) | 0.14 (+2 pp) | 0.61 (+8 pp) |
| | Qwen2.5 14B | Reasoning | 0.45 (+23 pp) | 0.38 (+19 pp) | 0.55 (+28 pp) |
| | Qwen2.5 32B | Reasoning | 0.55 (+16 pp) | 0.27 (+8 pp) | 0.41 (+12 pp) |

## I.6. Extreme Low-Information Elicitation

Tables 8 and 9 demonstrate that latent capabilities can be surfaced with remarkably little information. We highlight several notable findings:

**Single-example fine-tuning.** For Llama models on arithmetic tasks, fine-tuning on a single randomly sampled example—containing fewer than 10 bits of task-specific information—yields substantial accuracy improvements: +96 pp for Llama 3.1 8B on addition/subtraction, +66 pp for Llama 3.2 1B. In contrast, TinyStories–1B shows 0 pp improvement under identical conditions, as the capability must be taught rather than elicited.

**Few-parameter fine-tuning.** Training only 10 randomly sampled parameters on the full dataset yields comparable gains: Llama 3.1 8B achieves 91% accuracy (+91 pp) on addition/subtraction with 10 parameters. This confirms that the information bottleneck, not the parameter count per se, determines elicitation efficacy.

**Information content.** A single arithmetic example (e.g., "What is 23 + 45? 68") contains approximately 3–7 bits of answer information ($\log_2$ of the answer space). That such minimal information suffices to unlock near-perfect performance implies the model already possesses the computational capability; fine-tuning merely provides a "pointer" to activate it.

These findings have direct safety implications: if dangerous capabilities are latent, the barrier to elicitation may be far lower than dataset size alone would suggest.

## I.7. Comparison to Instruction-Tuned Models

Table 10 compares the zero-shot performance of instruction-tuned (IT) Llama models to the performance of base models fine-tuned on minimal data. Despite achieving 0% zero-shot accuracy, base models fine-tuned on very few examples consistently outperform the corresponding instruction-tuned models, which have undergone extensive RLHF and instruction tuning. This demonstrates that low-information elicitation via supervised fine-tuning can unlock capability that post-training alignment procedures do not fully surface.

These results also clarify why we report 0% zero-shot accuracy for base models (Table 9): this reflects the models' inability to follow instructions and produce correctly formatted responses, not an absence of the underlying capability. The capability is latent and can be surfaced with minimal information, often surpassing what instruction tuning achieves.

## I.8. Random Initialization as a Teaching Baseline

For completeness, we perform the same fine-tuning experiments on randomly initialized Llama 3.2 1B architectures (no pretraining) TinyStories-v2 (Tables 5 and 7). This provides an additional teaching baseline where *all* capability must be learned from scratch, as no preexisting representational structure exists. For randomly initialized Llama 3.2 1B model variants, we observe non-monotonic EDL learning dynamical signatures and capacity limits EDL/$P^*$ > 1 bit/parameter, which are consistent with EDL signatures and capacity limits observed in other teaching settings.

## I.9. Few-Shot Prompting Baselines

Table 11 reports few-shot prompting performance as an independent baseline for validating elicitation classification. Significant improvement under few-shot prompting (without

*Table 9.* Performance after full fine-tuning on a single randomly sampled example. Columns show zero-shot accuracy, 1-example accuracy, and absolute improvement ($\Delta$). Large $\Delta$ with a single example indicates elicitation of latent capability; minimal $\Delta$ indicates the capability must be taught. For binary/multiple choice tasks, we report balanced accuracy to account for class imbalances.

| Model | Addition/Subtraction | | | Multiplication | | | BoolQ | | | ARC–Easy / Challenge | | |
|---|---|---|---|---|---|---|---|---|---|---|---|---|
| | 0-shot | 1-ex | $\Delta$ | 0-shot | 1-ex | $\Delta$ | 0-shot | 1-ex | $\Delta$ | 0-shot | 1-ex | $\Delta$ |
| Llama 3.2 1B | 0.0 | 0.66 | 0.66 | 0.0 | 0.31 | 0.31 | 0.5 | 0.61 | 0.11 | 0.0 / 0.0 | 0.65 / 0.27 | 0.65 / 0.27 |
| Llama 3.2 3B | 0.0 | 0.86 | 0.86 | 0.0 | 0.54 | 0.54 | 0.5 | 0.62 | 0.12 | 0.61 / 0.25 | 0.74 / 0.52 | 0.13 / 0.27 |
| Llama 3.1 8B | 0.0 | 0.96 | 0.96 | 0.0 | 0.67 | 0.67 | 0.5 | 0.74 | 0.24 | 0.73 / 0.48 | 0.87 / 0.69 | 0.15 / 0.21 |
| TinyStories 1B | 0.0 | 0.0 | 0.0 | 0.0 | 0.0 | 0.0 | 0.0 | 0.5 | 0.5 | 0.0 / 0.0 | 0.22 / 0.24 | 0.22 / 0.24 |

*Table 10.* Comparison of instruction-tuned (IT) model zero-shot performance to base model performance after minimal supervised fine-tuning (SFT). Base models achieve 0% zero-shot but outperform IT models after fine-tuning on very few examples. $n^*$ denotes the approximate number of SFT examples at which the base model surpasses the IT model's zero-shot performance. All accuracy values in %.

| Model | Addition/Subtraction | | | Multiplication | | |
|---|---|---|---|---|---|---|
| | IT 0-shot | Base 1-ex | $n^*$ | IT 0-shot | Base 1-ex | $n^*$ |
| Llama 3.2 1B | 20.2 | 65.7 | 1 | 17.6 | 30.9 | 1 |
| Llama 3.2 3B | 41.3 | 85.9 | 1 | 35.8 | 54.0 | 1 |
| Llama 3.1 8B | 56.1 | 96.0 | 1 | 45.3 | 67.0 | 1 |

| Model | BoolQ | | | ARC–Easy / Challenge | | |
|---|---|---|---|---|---|---|
| | IT 0-shot | Base 1-ex | $n^*$ | IT 0-shot | Base 1-ex | $n^*$ |
| Llama 3.2 1B | 71.5 | 64.9 | 8 | 64.1 / 37.9 | 61.5 / 35.0 | 3 / 7 |
| Llama 3.2 3B | 73.7 | 74.4 | 1 | 70.0 / 45.1 | 71.3 / 46.7 | 1 / 1 |
| Llama 3.1 8B | 84.6 | 83.4 | 3 | 77.8 / 54.9 | 81.0 / 55.1 | 1 / 1 |

parameter updates) provides evidence that the capability is latent, consistent with EDL's classification of these tasks as elicitation-dominated.

These results independently validate EDL's classification of learning regimes, using an elicitation technique that involves no parameter updates. Llama models show substantial few-shot improvements on all arithmetic tasks (up to 68 pp on addition/subtraction and 52 pp on multiplication for Llama 3.1 8B), consistent with EDL's classification of these tasks as elicitation-dominated. TinyStories–1B base and format-only pre-taught variants show zero improvement under few-shot prompting on all tasks, consistent with EDL's classification of these as teaching-dominated: the capability is absent and cannot be surfaced without parameter updates.

Notably, the pre-taught TinyStories variants exhibit nontrivial few-shot improvements *only* on the specific operation that was pre-taught: pre-teaching addition/subtraction enables few-shot elicitation of addition/subtraction (11.9% at 16 shots) but not multiplication (0%), and vice versa. This mirrors the OOD pre-teaching controls observed in EDL signatures and capacity thresholds (Table 6), where only in-distribution pre-teaching converts the task from teaching to elicitation. The consistency between these two independent methods—one information-theoretic (EDL), the other behavioral (few-shot prompting)—strengthens the evidence that EDL signatures reflect genuine differences in latent capability rather than artifacts of training dynamics.

*Table 11.* Few-shot prompting accuracy (%) for Llama and TinyStories base models using $k = 0, 16$ shots. Pre-teach format, add/sub, and mult refer to TinyStories 1B models which have been pre-taught output format or the relevant operation using operator notation, respectively. Zero and few-shot evaluations reported here all use examples expressed in natural language. Significant improvement over zero-shot (0-shot) provides independent evidence of latent capability.

| Model | Add/Sub | | Mult | |
|---|---|---|---|---|
| | 0-shot | 16-shot | 0-shot | 16-shot |
| Llama 3.2 1B | 0.0 | 29.0 | 0.0 | 13.6 |
| Llama 3.2 3B | 0.0 | 58.9 | 0.0 | 42.1 |
| Llama 3.1 8B | 0.0 | 67.7 | 0.0 | 51.9 |
| TinyStories 1B | 0.0 | 0.0 | 0.0 | 0.0 |
| Pre-teach format | 0.0 | 0.0 | 0.0 | 0.0 |
| Pre-teach add/sub | 2.0 | 11.9 | 0.0 | 0.0 |
| Pre-teach mult | 0.0 | 0.0 | 1.4 | 8.7 |

## I.10. Random Labels: EDL is Negligible When No Generalizable Information Exists

EDL could in principle measure artifacts of the training process rather than genuine capability-relevant learning. To validate that EDL measures learnable structure, we conduct random label controls.

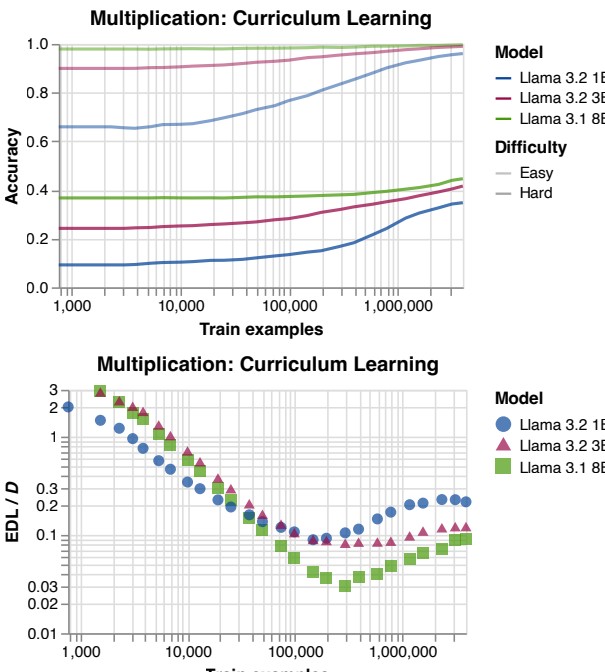

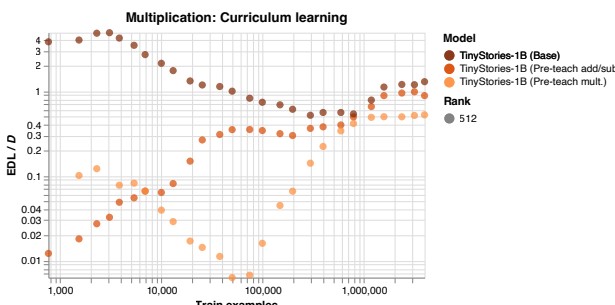

*Figure 9.* **Curriculum learning reveals the transition from elicitation to teaching as problem difficulty increases.** EDL per token versus dataset size for TinyStories-1B base (brown), pre-taught on easy multiplication (dark orange), and pre-taught on addition/subtraction, then fine-tuned on progressively harder problems. Easy problems (<10K examples): elicitation signatures (decreasing EDL/$D$). Hard problems (>100K examples): teaching signatures (increasing EDL/$D$).

*Figure 8.* **Performance saturates on easy problems before hard problems, with teaching signatures corresponding to improvements on harder problems.** Accuracy by problem difficulty level for Llama 3.2 1B fine-tuned on multiplication. *Top*: Performance on easy problems (light) saturates early during the elicitation-dominated phase; hard problems (dark) improve later during the teaching-dominated phase. *Bottom*: EDL per token crosses from decreasing (elicitation) to increasing (teaching) at the dataset size where hard-problem accuracy begins to improve.

### I.10.1. DESIGN

We replace training labels with random permutations, breaking the correspondence between inputs and outputs while preserving marginal statistics. If EDL measures generalizable information, it should collapse to near zero under this manipulation, as there is nothing to generalize.

We consider two conditions: (1) fully random labels independent between train and test, and (2) fixed random permutation (same mapping for train and test, but semantically arbitrary). Condition (1) tests whether EDL detects absence of any structure; condition (2) tests whether EDL detects learnable-but-arbitrary structure.

### I.10.2. RESULTS

Figure 10 shows EDL versus train set size for original labels compared to randomly permuted labels.

Models exhibit low, flat EDL under random labels—around 1,000–3,000 bits for arithmetic tasks regardless of dataset size, compared to over 10 million bits with original labels

(TinyStories)—validating that EDL measures the amount of generalizable structure learned, rather than memorization of the train set. The small positive EDL likely reflects learning the output format and optimal inductive bias (produce a uniformly sampled random number), exclusive of learning the deterministic input-output relationship. Since there is no general arithmetic algorithm that predicts the correct labels (aside from knowing the random seed and generator), no additional predictive information can be extracted from the train data once the task format has been learned.

### I.10.3. INTERPRETATION

These results validate EDL as a measure of learnable, generalizable structure. When structure exists (original labels), EDL grows with the information content of the training set. When structure is absent (random labels), EDL remains near zero regardless of training effort.

## J. Proposed Quantitative Definitions for Elicitation and Teaching

We propose the following definitions to precisely characterize elicitation and teaching both as learning *processes* and as learning *regimes*.

### J.1. Learning Processes

**Elicitation (process).** Elicitation as a learning process is characterized by decreasing learning efficiency with additional data. Elicitation surfaces existing knowledge by accessing information that has already been learned and stored in the model parameters. Formally, elicitation predominates

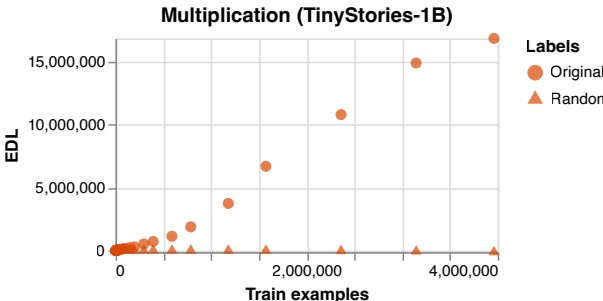

*Figure 10.* **EDL is constant with dataset size and negligible when no information generalizes.** EDL versus training examples for original labels (circles) versus randomly permuted labels (triangles). EDL monotonically increases with dataset size when fine-tuning on the original multiplication task labels (correct answers to multiplication problems). When labels are independently and randomly permuted within each of the train and test sets (removing all predictive algorithmic relationships between the data and corresponding labels), EDL remains constant as dataset size scales, indicating that the model gains no information from the train data that generalizes to the test set. All experiments shown use LoRA rank 512.

when:

$$\frac{\partial}{\partial n}\left(\frac{\text{EDL}(n)}{n}\right) < 0$$

where $n$ is the number of training examples and $\text{EDL}(n)$ is the excess description length at dataset size $n$.

**Teaching (process).** Teaching as a learning process is characterized by increasing learning efficiency with additional data (up to a saturation point). Teaching builds new capability by direct instruction when the capability cannot be deduced from existing knowledge. Formally, teaching predominates when:

$$\frac{\partial}{\partial n}\left(\frac{\text{EDL}(n)}{n}\right) > 0$$

### J.2. Learning Regimes

**Elicitation (regime).** Elicitation as a learning regime is characterized by limited information required to achieve maximum capability. The parameter capacity threshold is low ($P^* \approx 0.01$–$0.1$ bits/parameter), reflecting that minimal new information must be encoded.

**Teaching (regime).** Teaching as a learning regime is characterized by substantial information required because minimal relevant capability exists originally. The parameter capacity threshold is high ($P^* \approx 1$ bit/parameter), reflecting that significant new information must be encoded.

### J.3. Relationship Between Learning Processes and Regimes

The process (what happens at each $n$) and regime (overall information requirements) are related but distinct:

- A model in the *elicitation regime* will predominantly exhibit *elicitation process* signatures across most dataset sizes.

- A model in the *teaching regime* will initially exhibit *teaching process* signatures (increasing EDL/token), followed by *elicitation process* signatures once the capability is learned or parameter capacity is saturated.

- The crossover from teaching to elicitation process reflects the point at which the model has acquired sufficient capability that additional data becomes primarily redundant rather than informative.

### J.4. Capacity Thresholds Under Finite Compute

Capacity thresholds $P^*$ mark where learning *efficiency* drops (*i.e.*, significantly slows or degrades relative to full fine-tuning)—not where learning stops entirely. With infinite compute, less-capable models eventually absorb more information (higher EDL) and learn more. However, with finite compute budgets, this relationship can reverse.

When a model lacks prerequisite knowledge, learning may stall before discovering efficient algorithms, causing it to memorize inefficiently and hit capacity limits at low accuracy. Pre-teaching can enable the model to learn compressive algorithms early, freeing parameter capacity for additional learning. The pre-taught model may then achieve both higher accuracy *and* higher fine-tuning EDL than the base model trained with equivalent compute.

This is not a contradiction: the base model's low capacity $\text{EDL}/P^*$ reflects inefficient learning that stalls, while the pre-taught model's low $\text{EDL}/P^*$ reflects efficient learning that continues. We report absolute performance alongside capacity metrics throughout to distinguish these cases.

