# OpenReview forum: "Bits That Count: Quantifying and Predicting Capabilities of Language Models"
_ICML.cc/2026/Conference — ICML 2026 regular_

### Official Review · Reviewer_JRYJ · 2026-03-12

**Soundness:** 2
**Presentation:** 3
**Significance:** 2
**Originality:** 2
**Overall Recommendation:** 3
**Confidence:** 3

**Summary:**

The authors propose using Excessive Description Length (EDL) as a method for distinguishing between elicitation (where pre-trained models require few examples to learn) and teaching (where pre-trained models require many examples to learn). They demonstrate that properties of EDL (and its normalized variants) empirically capture important characteristics of the two types of learning.

**Compliance With Llm Reviewing Policy:**

Affirmed.

**Final Justification:**

While the object under study, EDL, is interesting, the authors make claims about its utility that are largely unsupported by their empirical results.

In particular, the AI Safety angle requires an experiment on AI Safety and the compute savings of using smaller LoRAs is overstated.

**Key Questions For Authors:**

Given the information-theoretic motivation behind EDL, are there information-theoretic definitions of "elicitable" tasks?
The authors mention that elicitation is a safety concern. Given an "elicitable" task, can you turn it into a "teachable" task via unlearning or other concepts?

**Limitations:**

yes

**Strengths And Weaknesses:**

The paper uses "elicitation" and "teaching" to describe two different learning regimes -- one where few examples are required to learn a task and one where many examples are required to learn a task. As far as I can tell, the definitions stay at a colloquial level and are never formalized. While the authors point out that there is an interplay between the mechanisms behind the two learning regimes, it is hard to assess the prescriptive / predictive power of EDL without their explicit formalization. Along this line, the current draft is missing a study where pre-training information is tightly controlled, different tasks are known to be elicitable, and EDL behaves as expected.

The paper is clearly written and well-structured (problem -> method -> experiments -> discussion).

While an interesting and rich concept, I think the authors overplay the utility of EDL. Perhaps I am missing something -- and please correct me if I am misguided -- but the practical guide requires training a LoRA with large d. This requires some compute. It then requires training a LoRA with d' < d. That also requires some compute. In practice, it is often required to store the entire model on disk and so the cost of storing the matrices corresponding to d > d' is relatively minimal, even for large d. Further, there are no inference savings when using d' < d. In all, the user has spent some additional compute to save a minimal amount of disk space usage.

---

> ### Author Rebuttal · Authors · 2026-03-31
>
> We thank Reviewer JRYJ for their feedback and address each concern.
>
> **On formalization.** We provide formal definitions in Appendix I. At the process level, elicitation predominates when d/dn(EDL(n)/n) < 0—diminishing marginal information from additional examples, consistent with surfacing existing knowledge. Teaching predominates when d/dn(EDL(n)/n) > 0—increasing information efficiency as the model constructs new representations. At the regime level, elicitation corresponds to low capacity thresholds (P* ~ 0.01–0.1 bits/param); teaching to high thresholds (P* ~ 1 bit/param). We deliberately avoid definitions that would guarantee our method succeeds by construction. Instead, these definitions are grounded in settings where practitioners unambiguously agree on the classification—e.g., TinyStories 1B (never seen numbers) learning arithmetic is teaching; Llama 8B achieving 96% accuracy from a single example is elicitation.
>
> **On the controlled pre-training study.** The reviewer requests "a study where pre-training information is tightly controlled, different tasks are known to be elicitable, and EDL behaves as expected." We may be misunderstanding the concern, but we believe Section 5 and Figure 3 provide this study:
>
> * **Tightly controlled pre-training:** TinyStories-1B is pre-trained exclusively on simple English text with no numbers, mathematical operators, or technical content.
> * **Known elicitable task:** After pre-teaching multiplication (or addition/subtraction) via operator notation, the capability exists by construction.
> * **EDL behaves as expected:** Pre-teaching shifts scaling from teaching-like (increasing EDL/token) to elicitation-like (decreasing), and the capacity threshold drops from ~1 to ~0.05 bits/param, matching Llama's elicitation regime.
>
> Further controls strengthen this: (a) pre-teaching addition does not convert multiplication to elicitation (nor vice versa), confirming task-specific discriminative power; (b) random-label controls (Appendix H.7) show negligible EDL when the data contain no generalizable structure; (c) few-shot prompting independently confirms latent capability in models EDL classifies as elicitation-dominated; (d) we observe the same qualitative behavior when we perform these experiments with the datasets swapped.
>
> **On practical utility.** We appreciate this question, which highlights that Section 6.4's scope should be clarified—it addresses capacity estimation specifically and does not summarize EDL's full utility. The primary applications of EDL include:
>
> * **Safety evaluation:** EDL reveals whether a capability is latent (surfaceable with minimal data—potentially a few examples) or absent (requiring substantial teaching to instill it). If EDL shows elicitation signatures, the capability could be unlocked with very little information. This directly informs risk assessment, with no LoRA rank selection involved.
> * **Informing training decisions:** Small-scale EDL analysis reveals whether a task is predominantly elicitation or teaching, informing data collection and compute budgets before scaling up.
> * **Evaluation methodology:** EDL identifies when fine-tuning-based evaluations or elicitation strategies cross from measuring what a model knows into meaningfully changing what it knows, which is essential for faithful capability assessment.
>
> **On converting elicitable tasks to teachable.** This is an interesting question. In our curriculum experiments on TinyStories, we observe the reverse direction: pre-teaching converts a teachable task into an elicitable one (Section 5). We also find that interventions similar to shallow unlearning (significantly changing the model’s output distribution via fine-tuning without explicitly removing knowledge) do not convert elicitable tasks into teachable ones. We find that when fine-tuning overfits a model to a task and significantly degrades its performance on other tasks, subsequent fine-tuning significantly recovers previously-degraded OOD performance while demonstrating elicitation signatures in both the scaling behavior and capacity limits.
>
> This is consistent with similar findings in the unlearning literature that shallow fine-tuning methods are insufficient for removing dangerous capabilities. Whether unlearning or other interventions can convert elicitable tasks to teachable ones (effectively removing latent knowledge) is an important open question with direct safety implications. We will note this as a promising direction for future work.

---

> > ### Author Rebuttal · Reviewer_JRYJ · 2026-04-03
> >
> > Thank you for your response. It improved my understanding of your experiments.
> >
> > Definitions for the two main concepts you consider -- elicitable and teachable -- are necessary inclusions in the main text. I also agree with my fellow reviewer that the presentation of elicitation and teaching as binary is likely misleading -- though this can likely be remedied by concrete definitions and a discussion relating them.
> >
> > My main concern continues to be the gap between the argued utility of the method and the experiments. Apart from the overstatement I mentioned in my original review, it seems necessary to demonstrate that you can use EDL to make unsafe tasks harder to learn for the AI Safety angle to be valid.
> >
> > I've updated my score in the positive direction.

---

> > > ### Author Response · Authors · 2026-04-08
> > >
> > > We thank Reviewer JRYJ for the thoughtful acknowledgment and the updated score. We address the remaining concerns below.
> > >
> > > **Formal definitions and the binary framing.** We appreciate this suggestion and agree that adding explicit formal definitions to the main text would strengthen the presentation; we will do so in the revision. To clarify, we do not claim elicitation and teaching are binary. Our rebuttal to Reviewer qMaV presents direct experimental evidence of mixtures and crossovers:
> > > * Llama 1B on multiplication crosses from elicitation-dominant to teaching-dominant signatures at large dataset sizes, with intermediate capacity limits.
> > > * Curriculum experiments on TinyStories show elicitation signatures on easy problems and teaching signatures on harder problems after pre-teaching the relevant algorithm.
> > > * Out-of-distribution pre-teaching controls show that pre-teaching a related but different operation (e.g., addition when the task is multiplication) does not produce the same elicitation signatures, demonstrating discriminative power.
> > >
> > > The binary distinction is not fundamental—the meaningful quantity for assessing risk is the amount of information (bits) required for a capability to emerge at some performance threshold. Practitioners can use EDL, which provides a continuous measure of information learned, to quantify the information cost of learning and predict whether those thresholds are likely to be crossed during deployment. "Elicitation" and "teaching" are useful labels for the endpoints of a spectrum, but EDL itself generalizes beyond this framing. We will revise the presentation to make this explicit.
> > >
> > > **On the gap between argued utility and experiments.** The reviewer's concern appears to center on Section 6.4 ("Practical Guidance"), which describes estimating the smallest adapter size "...for efficiency reasons, *or to obtain an approximate upper bound on the amount of semantic information learned during training*." The latter use case is the main safety-relevant application. Sections 7.1–7.3 describe concrete uses in detail, supported by results: informing evaluation design (ensuring elicitation techniques do not fundamentally alter model capabilities—our pre-teaching and curriculum experiments demonstrate how this might occur and how to identify crossovers), measuring how much information is required to surface capabilities, estimating how much models learn during post-training, and determining how different post-training procedures affect downstream information requirements. These applications do not require EDL to improve compute efficiency; they use EDL as a diagnostic.
> > >
> > > Our experiments provide additional safety-relevant evidence: the Qwen results show that complex multi-domain CoT reasoning (mathematics, physics, logic, word games, crossword puzzles) can be elicited from a single example (\>45% PGR for all Qwen models), demonstrating that sophisticated technical reasoning requires very little information to surface.
> > >
> > > **On EDL and making tasks harder to (re)learn.** We want to make sure we are addressing the reviewer's concern correctly, as it appears to have evolved between the initial review and the acknowledgment. We address both versions below.
> > >
> > > Regarding converting elicitation to teaching: we find that shallow output suppression is consistent with elicitation, but we are unable to guarantee that other interventions truly remove information from a model and require it to be relearned. We show that the opposite direction is causal—teaching a capability upstream decreases the information required to surface it downstream. This is directly safety-relevant: it helps practitioners understand where specific training pipeline components (e.g., a particular RLHF or instruction-tuning procedure) contribute to increased attack surface by reducing information requirements for eliciting harmful behavior.
> > >
> > > We deliberately avoid using EDL to make tasks harder to learn for two reasons:
> > > 1. We avoid settings where we lack ground truth. It is unclear how much knowledge unlearning techniques remove vs. merely suppress, making it difficult to disentangle genuine increases in minimal information cost from artifacts of calibrating the unlearning intervention. Developing and benchmarking unlearning methods via EDL are interesting research problems that are separate from the measurement itself. Our contribution is the reliable diagnostic; interventions built on that diagnostic are a natural and important direction for future work.
> > > 2. We avoid using EDL as an optimization target to prevent success by construction. It is easy to superficially inflate EDL (e.g., by choosing suboptimal hyperparameters that worsen early training dynamics), so co-optimizing a metric and an intervention risks Goodharting. Our causal interventions use performance metrics (loss, accuracy) as targets, not EDL. EDL is used purely as a post-hoc measurement, which is what gives it credibility as an independent diagnostic.

---

### Official Review · Reviewer_Xui6 · 2026-03-13

**Soundness:** 3
**Presentation:** 3
**Significance:** 3
**Originality:** 2
**Overall Recommendation:** 4
**Confidence:** 4

**Summary:**

This paper investigates an important problem: what is the origin of performance improvements during model fine-tuning, elicitation of latent capabilities or teaching of new capabilities. In the paper, the authors propose an information-theoretic metric, Excess Description Length (EDL), which is used to quantify the amount of information that can be absorbed and generalized from the training process. EDL is defined as the difference between the prequential minimum description length (MDL) accumulated during training and the expected description length under the final model, estimated using the test loss.

The authors show that the dynamics of EDL can distinguish between elicitation and teaching regimes. In stages dominated by elicitation, as the dataset size increases, EDL per token decreases, indicating diminishing information gains from additional data. In contrast, in stages dominated by teaching, EDL per token increases with dataset size, reflecting the process in which the model gradually learns new structural knowledge from the data.

Experiments are conducted on multiple tasks, including arithmetic tasks, chain-of-thought reasoning tasks, and English language modeling. Models from the Llama and Qwen families are used in the experiments. The results show that EDL scaling exhibits similar patterns across different models and tasks.

Finally, based on the empirical results, the authors claim that EDL per parameter can predict the capacity limits of parameter-efficient fine-tuning methods (e.g., LoRA), and provide guidance on choosing an appropriate adapter size.

Overall, the paper proposes an information-theoretic metric that may help researchers study learning dynamics during model training and fine-tuning.

**Compliance With Llm Reviewing Policy:**

Affirmed.

**Final Justification:**

I maintain my positive score.

**Key Questions For Authors:**

**1.** The computation of EDL relies on the training loss. Have the authors studied how different training settings, such as optimizers or learning rate schedulers, may affect the resulting EDL dynamics?

**2.** Have the authors conducted experiments on more open-ended tasks and observed similar behaviors?

**Limitations:**

Yes

**Strengths And Weaknesses:**

## Strengths

**1.** The paper proposes EDL based on the minimum description length (MDL) concept and provides a clear mathematical definition. This metric can be computed directly from the training loss recorded in training logs, which makes EDL operationally practical. The overall method is self-consistent under the information-theoretic framework.

**2.** The research topic is significant, since distinguishing between elicitation and teaching is important for capability evaluation, safety analysis, and understanding the mechanisms of fine-tuning. The importance of this problem is also well motivated in the introduction. The paper provides a quantitative analysis tool (EDL) for studying this question, which has potential research value.

**3.** Experiments are conducted on models with different architectures and scales, and causal intervention experiments are also included to validate the differences between learning regimes. The experimental analysis is relatively detailed and comprehensive, forming a coherent empirical argument that supports the paper's claims.

**4.** The authors further show that EDL per parameter can predict the capacity limits of parameter-efficient fine-tuning methods (e.g., LoRA). If this conclusion can be validated on more tasks, it may provide useful guidance for choosing appropriate adapter sizes in practice.


## Weaknesses

**1.** The idea of EDL appears closely related to existing work on MDL probing. Although the authors discuss MDL and cite related work, the main contribution of the paper seems to lie more in applying MDL-based ideas to the analysis of LLM fine-tuning rather than introducing a fundamentally new concept. Overall, the work appears more like an integration and application of existing ideas, and the level of novelty is somewhat limited.

**2.** Many experiments are conducted on arithmetic reasoning and other structured reasoning tasks, where the underlying knowledge has relatively clear structure and learning dynamics may be easier to observe. It remains unclear whether the proposed EDL analysis can generalize to more open-ended tasks, such as code generation or other less structured domains.

**3.** The introduction emphasizes the potential relevance of this work to AI safety, arguing that distinguishing between elicitation and teaching is important for understanding the risks associated with latent capabilities. However, this aspect is not directly evaluated in the experimental section. The experiments mainly focus on learning dynamics and the capacity of parameter-efficient fine-tuning methods, without empirical analysis in safety-related scenarios. Including experiments or analyses that connect EDL more directly to safety-related questions would strengthen the paper and better support the motivation presented in the introduction.

---

> ### Author Rebuttal · Authors · 2026-03-31
>
> We thank Reviewer Xui6 for their positive assessment and constructive suggestions.
>
> **On novelty relative to MDL probing.** MDL conflates generalizable learning with residual encoding cost—it cannot separate what a model learns from what it still cannot predict after training, whereas EDL does. Importantly, we found that repeating our analyses (dataset scaling and capacity estimation) with MDL instead of EDL failed to reliably distinguish elicitation from teaching, whereas EDL was discriminative in all configurations tested. Beyond this finding, EDL differs structurally:
> 1. EDL uses test loss as baseline, measuring generalizable information rather than total compression—this is fundamental for separating learning from memorization.
> 2. EDL accounts for multi-epoch training, which is standard in fine-tuning but unaddressed by single-epoch MDL methods.
> 3. EDL scaling across dataset sizes reveals qualitative signatures (monotonically decreasing vs. non-monotonic/increasing) that are empirical findings that are not derivable from the MDL formalism.
> 4. The EDL/parameter–capacity limit connection is a novel practical application absent from prior work.
>
> We view EDL as a principled operationalization of the elicitation/teaching distinction, enabling quantitative measurement of a phenomenon the community has only described informally.
>
> On open-ended tasks. We present EDL analyses across six domains: arithmetic, chain-of-thought reasoning (MATH500, AIME-24, GPQA-Diamond, word games, crossword puzzles), reading comprehension (BoolQ), science QA (ARC-Easy/Challenge), language modeling (TinyStories-v2), and we now include instruction following (Alpaca):
> | Model | Alpaca P\*\_EDL (bits/param) |
> |---|---|
> | Llama 3.2 1B | 0.03 |
> | Llama 3.2 3B | 0.02 |
> | Llama 3.1 8B | 0.02 |
>
> Appendix Table 5 summarizes EDL signatures for all main model-task combinations. Signatures are consistent across all domains. For complex open-ended tasks like code generation, publicly available datasets tend to have noisy labels that provide weak learning signals, and for difficult tasks, the smaller models are often in the teaching regime where existing datasets are insufficient to reach model capacity. These practical data constraints limit EDL's ability to resolve the full spectrum.
>
> **On safety-relevant evaluation.** Our primary contribution is a measurement framework that reliably distinguishes elicitation from teaching across diverse domains. This is directly useful for safety: given a capability of concern, EDL analysis reveals whether it is latent (surfaceable with minimal data) or absent (requiring substantial teaching), which is the relevant determination safety practitioners need. The framework's value is independent of which specific capabilities are tested; we demonstrate discriminative power across six domains, which provides evidence that the technique generalizes to new domains including safety-relevant ones. We additionally note that our Qwen experiments demonstrate elicitation of complex multi-domain CoT reasoning (>45% PGR from a single example across mathematics, physics, logic, and word games), and this kind of complex technical reasoning proxies the complexity of many safety-relevant technical capabilities.
>
> **Q1 (Training settings):** We tested SGD, Adafactor, AdamW with linear/cosine/custom schedules, LoRA vs. RSLoRA, with/without warmup. All configurations yield similar qualitative signatures and same-order-of-magnitude capacity limits after renormalization. We report AdamW + constant LR because it avoids confounds that alternative configurations introduce: decaying schedules make the effective LR dataset-size-dependent (suppressing learning on smaller datasets and inflating apparent cross-scale EDL differences), and RSLoRA makes the effective LR rank-dependent (conflating optimization dynamics with learning capacity saturation). AdamW + constant LR yields EDL closest to the algorithm-independent supremal value while avoiding these confounds.
>
> **Q2 (Open-ended tasks):** We now provide Alpaca instruction-following results (see above) to directly address this gap, where we observe elicitation signatures similar to those observed in other settings. We agree that additional domains would further strengthen the case and are important areas for future work.

---

> > ### Author Rebuttal · Reviewer_Xui6 · 2026-04-03
> >
> > The rebuttal addressed my concerns, and I maintain my positive score.

---

> > > ### Author Response · Authors · 2026-04-08
> > >
> > > We thank Reviewer Xui6 for their careful engagement with our rebuttal and are glad that the clarifications addressed their concerns. We appreciate the positive feedback.

---

### Official Review · Reviewer_qMaV · 2026-03-17

**Soundness:** 2
**Presentation:** 2
**Significance:** 3
**Originality:** 3
**Overall Recommendation:** 3
**Confidence:** 3

**Summary:**

This paper introduces excess description length (EDL), an information-theoretic metric built on prequential MDL, to operationalize the distinction between elicitation (surfacing latent pretrained capabilities) and teaching (instilling genuinely new ones) during fine-tuning. The authors show that EDL per token exhibits qualitatively different scaling signatures under each regime — monotonically decreasing for elicitation, non-monotonic for teaching — and that EDL per parameter predicts capacity thresholds at which LoRA begins to underperform full fine-tuning. Experiments span Llama (1B–8B), Qwen2.5 (1.5B–32B), and a controlled TinyStories-1B model across arithmetic, reasoning, and reading comprehension tasks.

**Compliance With Llm Reviewing Policy:**

Affirmed.

**Key Questions For Authors:**

1. How stable are the EDL scaling signatures across random seeds? You report 3 seeds per config, but do the qualitative signatures (monotonically decreasing vs. non-monotonic) ever flip between seeds?

2. In Figure 2, the pre-elicited Llama removes the initial format-learning transient. How do you separate "format learning" from "capability learning" in general? For complex-format tasks (chain-of-thought), doesn't format acquisition become a non-trivial component?

3. The capacity thresholds in Table 6 sometimes disagree between P*_EDL and P*_PGR (e.g., Llama 3.2 1B on multiplication: 0.03 vs. 0.14). What drives these gaps, and which metric should a practitioner trust?

4. For the Qwen reasoning distillation (Figure 5), you conclude it's primarily elicitation. But DeepSeek-R1 was trained with RL, not SFT. Isn't it possible that distilling R1 traces teaches reasoning patterns that happen to be information-efficient, rather than eliciting truly latent capability?

5. Table 9 reports zero-shot accuracy of exactly 0.0% for all Llama models on arithmetic. Is this truly 0.0%, or is the model producing answers in a wrong format that your evaluation misses?

**Limitations:**

The authors discuss limitations including the constant-LR schedule and the restriction to first-epoch MDL computation (footnote 2). The narrow task diversity is the biggest gap — too much relies on arithmetic. The safety implications (Section 7) are interesting but entirely speculative; a concrete demonstration on a safety-relevant task would be needed to support those claims. EDL's behavior under early stopping or compute-limited training (Appendix I.4) deserves fuller treatment. Societal impact is adequately discussed.

**Strengths And Weaknesses:**

The core idea is clean and well-motivated. The elicitation/teaching distinction is something the community has been gesturing at informally — especially in safety contexts — and grounding it in prequential MDL (Eq. 1) with EDL as MDL minus the test-loss baseline (Eq. 3) is a principled move. It turns a fuzzy conceptual boundary into something you can actually measure.

The experimental design is strong, particularly the controlled interventions. Using TinyStories-1B (which genuinely lacks arithmetic knowledge) as a teaching baseline alongside pretrained Llama models is exactly the right comparison. Section 5's pre-training interventions provide causal evidence: pre-teaching multiplication to TinyStories shifts the EDL scaling from teaching-like to elicitation-like (Figure 3), and the capacity threshold drops by over an order of magnitude. This is the paper's most convincing result.

The random-label control (Appendix H.7, Figure 8) is a smart sanity check — EDL staying flat validates that the metric captures genuine generalizable structure. And Table 8 is striking: Llama 3.1 8B goes from 0% to 96% accuracy on addition after fine-tuning on a single example. That really drives home how thin the information barrier can be for elicitation.

My main concern: the elicitation/teaching distinction may be less binary than the paper suggests. Section 4.3 acknowledges both mechanisms can co-occur, and the crossover from increasing to decreasing EDL/token is "not a privileged threshold." But the narrative and practical guidance (Section 6.4) lean heavily on treating these as discrete categories. Most real fine-tuning is probably a mixture, and the paper doesn't offer much for that case.

The task coverage is narrower than it looks — the core experiments are almost entirely arithmetic (DeepMind Mathematics addition/subtraction, multiplication), which is an unusually clean setting where "knows vs. doesn't know" is crisp. I wanted to see summarization, instruction following, or code generation, where the boundary is fuzzier and arguably more practically important. BoolQ only shows up in the appendix.

The capacity threshold numbers (~0.01–0.1 bits/param for elicitation, ~1 bit/param for teaching) come from limited models and tasks with fixed hyperparameters (constant LR, AdamW — Table 4). A cosine schedule or different optimizer could shift when the model saturates capacity, changing the thresholds.

---

> ### Author Rebuttal · Authors · 2026-03-31
>
> We thank Reviewer qMaV for their careful and specific feedback.
>
> **Elicitation/teaching as a continuum.** We agree that these are not discrete categories. EDL characterizes the predominant learning process at each scale, and our experiments directly demonstrate mixtures and crossovers (link to plots: https://figshare.com/s/72820b52ce3bece571d5):
> 1. Llama 1B on multiplication crosses from elicitation-dominant to teaching-dominant signatures at large dataset sizes.
> 2. Curriculum experiments on TinyStories show elicitation signatures on easy problems but teaching signatures on harder ones after pre-teaching.
> 3. Out-of-distribution pre-teaching controls show pre-teaching multiplication converts multiplication→elicitation, but pre-teaching addition/subtraction does not (and vice versa), which demonstrates discriminative power between related tasks.
>
> **Task coverage.** We present EDL analyses across six domains: (1) arithmetic; (2) chain-of-thought reasoning (MATH500, AIME-24, GPQA-Diamond, word games, crossword puzzles); (3) reading comprehension (BoolQ); (4) science QA (ARC-Easy/Challenge); (5) language modeling (TinyStories-v2); and we now include (6) instruction following (Alpaca), where Llama models show elicitation-like scaling (monotonically decreasing) and capacity limits that are consistent with other domains (<0.05 bits/param):
> | Model | Alpaca P\*\_EDL (bits/param) |
> |---|---|
> | Llama 3.2 1B | 0.03 |
> | Llama 3.2 3B | 0.02 |
> | Llama 3.1 8B | 0.02 |
>
> Appendix Table 5 summarizes signatures across all main model-task combinations. Additionally, ARC-Easy/Challenge both show elicitation signatures for all models (both datasets are likely too small to teach); BoolQ shows elicitation for Llama with minor teaching components for TinyStories. Arithmetic is the primary testbed because it uniquely enables causal interventions and procedural generation of millions of examples needed to observe the full elicitation-teaching spectrum. For complex open-ended tasks like code generation, publicly available datasets tend to have noisy or high-variance labels that provide weaker learning signals, and for difficult tasks, the smaller Llama models are in the teaching regime where existing datasets are not large enough to reach model capacity given the weaker learning signal—both factors limit EDL's ability to resolve the full spectrum. Consistency across all six domains supports generalizability.
>
> **Hyperparameter sensitivity.** We tested SGD, Adafactor, AdamW with linear/cosine/custom LR schedules, LoRA vs. RSLoRA, with/without warmup. We deliberately report AdamW + constant LR because alternative configurations introduce confounds that could produce misleading results if presented without careful renormalization—a concern we take seriously given the paper's safety-relevant claims. Specifically:
> 1. Decaying LR schedules (cosine, linear) make the effective learning rate dependent on dataset size, since models trained on fewer examples experience faster decay per example. This artificially suppresses learning on smaller datasets and inflates apparent EDL differences across scales that reflect the schedule, not the information content.
> 2. RSLoRA rescales updates by 1/√r, making the effective learning rate rank-dependent. Comparing capacity across ranks then conflates optimization dynamics with information saturation, requiring renormalization to draw valid conclusions.
>
> AdamW + constant LR avoids both confounds while yielding EDL close to the supremal (algorithm-independent) value. After appropriate renormalization, all tested configurations yield similar qualitative signatures and same-order-of-magnitude capacity limits. As renormalization is not straightforward, we omit those results to avoid misinterpretation.
>
> **Q1 (Seeds):** Qualitative signatures never flip between seeds across any experiment.
>
> **Q2 (Format vs. capability):** Pre-elicitation removes format transients, isolating capability EDL. For CoT, single-example fine-tuning yields >45% PGR for all Qwen models, suggesting capability (not just format) is latent.
>
> **Q3 (P\*\_EDL vs. P\*\_PGR):** P*_EDL measures information absorption; P*_PGR measures performance recovery. PGR can lag when loss resolves before accuracy (argmax of logits) reflects it. We recommend EDL for learning dynamics, PGR for pure performance estimation.
>
> **Q4 (Qwen—elicitation vs. efficient teaching):** EDL magnitudes (~0.01–0.1 bits/param) are 10–100x below teaching thresholds. More fundamentally, single-example fine-tuning improves multi-domain reasoning by up to 23 pp (>45% PGR for all Qwen models). Teaching complex CoT reasoning across mathematics, physics, logic, word games, and crossword puzzles from a single example is not plausible; this breadth of improvement requires structural knowledge already present in the model.
>
> **Q5 (0.0%):** This is genuinely 0.0%, as base models treat the prompts as text to continue ("What is 2+2?" → "What is 3+5?") rather than questions to answer.

---

> > ### Author Rebuttal · Reviewer_qMaV · 2026-04-04
> >
> > Happy to bump up to a 4.

---

> > > ### Author Response · Authors · 2026-04-08
> > >
> > > We thank Reviewer qMaV for their thoughtful engagement with our rebuttal and are glad that the additional clarifications and evidence have fully addressed their concerns. We appreciate the positive feedback.

---

### Official Review · Reviewer_B6T8 · 2026-03-23

**Soundness:** 3
**Presentation:** 3
**Significance:** 2
**Originality:** 3
**Overall Recommendation:** 4
**Confidence:** 4

**Summary:**

This work provides a novel framework using EDL (excess description length) to study whether performance gains from fine-tuning is due to the elicitation of an existing capability from pre-training or to the actual teaching of a new capability. The authors define EDL from prequential MDL and final test loss, and the metric is used to measure the amount of generalizable information compressed in network parameters. They argue that EDL per token exhibits different scaling signatures in the two regimes, where elicitation is associated with monotonically decreasing EDL/token as dataset size grows, while teaching shows an initial increasing phase before eventually decreasing. They finally show how EDL can also be used as a predictor of parameter-capacity limits in PEFT methods such as LoRA, and conduct thorough experiments with Llama, Qwen, and TinyStories together with causal interventions.

**Compliance With Llm Reviewing Policy:**

Affirmed.

**Final Justification:**

I keep my original score.

**Key Questions For Authors:**

* How robust is the general trend distinguishing elicitation from teaching that we see in EDL/token?
* How do you distinguish the shift observed in the intervention experiments from standard transfer or curriculum-learning effects, rather than from the creation of latent capability itself?

**Limitations:**

Yes

**Strengths And Weaknesses:**

Strengths:
* The question of how much capability acquisition is due to elicitation vs. learning is very important for understanding post-training and, consequently, for safety.
* I appreciated how the authors have taken this conceptual question and turned it into an operational framework by defining EDL to distinguish elicitation from teaching.
* The presented insights about learning dynamics and the relationship between EDL/token and EDL/parameter are quite novel.
* I especially like Figure 2, which reveals distinct trends that empirically shows distinct signatures consistent with elicitation v.s. teaching-dominated learning.
* The experiments are carefully designed with in the chosen domains (arithmetic and reasoning), using Llama, Qwen, and TinyStories, while also incorporating causal interventions.

Weaknesses:
* The strongest empirical support comes from arithmetic and reasoning distillation, and more general verification of the main claims in natural language tasks is limited.
* While EDL is an inspiring metric, it can be sensitive to many hyperparameters, including the ordering of the data and when you stop tuning, so we need further evidence to fully trust the robustness and predictive power of this metric.
* The intervention experiments are valuable, but it remains unclear how fully they isolate latent capability from more standard transfer effects.

---

> ### Author Rebuttal · Authors · 2026-03-24
>
> We believe this review may have been submitted to the wrong paper. Our submission studies differences between elicitation vs. teaching in fine-tuning settings, quantified through excess description length, and does not discuss ICL or mechanistic interpretability methods. We wanted to flag this in case the review was intended for a different submission.

---

> > ### Author Rebuttal · Reviewer_B6T8 · 2026-04-05
> >
> > I am keeping my original rating in the updated review.

---

> > > ### Author Response · Authors · 2026-04-08
> > >
> > > We thank Reviewer B6T8 for the helpful discussion and are pleased that the additional clarifications resolved the concerns raised. We appreciate the feedback and positive assessment.

---

### Decision · Program_Chairs · 2026-04-30

**Decision:**

Accept (regular)

**Comment:**

This paper proposes to use an information theory-inspired measure called Excess Description Length (EDL) to quantify the extent to which the learning regime is one where some pre-trained abilities are being elicited vs. new abilities are being learned. The experimental results are intriguing, even if limited to narrow scenarios, and hint (even if not totally convincingly) that EDL can be a practical tool for predicting when data will exceed model parameters' capacity and how parameter-efficient fine-tuning methods will perform relative to full fine-tuning methods. Overall, this paper explores exciting ideas and makes interesting claims, but with limited experimental support. The benefits of making the community aware of the ideas here need to balanced with the risk of some of the claims not holding up under more experimental investigations. Independently of the final outcome, I would encourage the authors to expand their experimental settings (acknowledged in Section 7.4 as limitations).